# Nucleolar protein TAAP1/*C22orf46* confers pro-survival signaling in non-small cell lung cancer

Marietta Döring[1], Melanie Brux[1,2], Maciej Paszkowski-Rogacz[2] ●, Pedro M Guillem-Gloria[3] ●, Frank Buchholz[1,2,4] ●, M Teresa Pisabarro[3], Mirko Theis[1,2] ●

**Tumor cells subvert immune surveillance or lytic stress by harnessing inhibitory signals. Hence, bispecific antibodies have been developed to direct CTLs to the tumor site and foster immune-dependent cytotoxicity. Although applied with success, T cell–based immunotherapies are not universally effective partially because of the expression of pro-survival factors by tumor cells protecting them from apoptosis. Here, we report a CRISPR/Cas9 screen in human non-small cell lung cancer cells designed to identify genes that confer tumors with the ability to evade the cytotoxic effects of CD8[+] T lymphocytes engaged by bispecific antibodies. We show that the gene *C22orf46* facilitates pro-survival signals and that tumor cells devoid of *C22orf46* expression exhibit increased susceptibility to T cell–induced apoptosis and stress by genotoxic agents. Although annotated as a non-coding gene, we demonstrate that *C22orf46* encodes a nucleolar protein, hereafter referred to as "Tumor Apoptosis Associated Protein 1," up-regulated in lung cancer, which displays remote homologies to the BH domain containing Bcl-2 family of apoptosis regulators. Collectively, the findings establish TAAP1/*C22orf46* as a pro-survival oncogene with implications to therapy.**

## Introduction

Over the past decade, advances in the understanding of the intricacies of the immune system have resulted in practice-changing innovation in the field of immuno-oncology (Chen et al, 2013; Watson et al, 2020). Although dysregulation of the immune system by tumor cells is a multifaceted and complex process, therapeutic approaches exploiting the exquisite anti-cancer capabilities of immune effector cells have seen unprecedented success in the clinic (Ribas & Wolchok, 2018; Cappell & Kochenderfer, 2023). The most widely applied immunotherapies harness the activities of CTLs (Cappell & Kochenderfer, 2023), fostering their anti-tumor

functions by inhibiting negative regulators such as immune checkpoint proteins (Ribas & Wolchok, 2018). Although applied successfully in many tumor entities (Hirsch et al, 2019), only a small proportion of non-small cell lung cancer (NSCLC) patients exhibit long-term responses to inhibitory checkpoint therapies (Vokes et al, 2023). The objective response rates are limited to about 25% (Hirsch et al, 2019), whereas subgroups such as epidermal growth factor receptor (EGFR) or ALK-mutated NSCLC patients hardly benefit from the treatment (Koyama et al, 2016). NSCLC is the most frequent subtype of lung carcinomas (Molina et al, 2008), providing the number one cause of cancer-related deaths globally (Bray et al, 2018), demanding alternative therapeutic concepts. In addition to therapies based on immune checkpoint blockade, other strategies evoking anti-cancer immunity have been developed, such as bispecific antibodies (biAbs) (Runcie et al, 2018). These molecules work by binding to TCR complex proteins such as CD3 and a common tumor antigen, thus engaging tumor and T cells (Huehls et al, 2015). Hence, they elicit a polyclonal CTL response that is not restricted by TCR specificity and surface presentation of tumor-peptide/MHC class I complexes (Carrasco-Padilla et al, 2022). Although biAbs efficiently elicit an immune response against cancer cells in vitro (Deisting et al, 2015) and in vivo (Zhao et al, 2019), their clinical application is often limited, partially because of the expression of pro-survival factors protecting tumor cells from T cell–induced apoptosis (Tuomela et al, 2022; Kaloni et al, 2023). For example, loss of the pro-apoptotic Bcl-2 family member Bid has been reported to reduce the sensitivity of cancer cells to granzyme B secreted by CTLs (Waterhouse et al, 2005). The identification of apoptosis regulators may hence facilitate a strategy to increase tumor susceptibility to T cell–induced killing and advance immunotherapies.

Loss-of-function genetic screening using CRISPR/Cas9 has been established as a powerful tool for the identification of novel gene functions implicated in various disease-related processes (Patel et al, 2017; Sreevalsan et al, 2020). Although these screens delivered valuable information on the genetic repertoire involved, the picture of resistance mechanisms and pro-survival signaling in tumor cells

[1]National Center for Tumor Diseases/University Cancer Center (NCT/UCC): German Cancer Research Center (DKFZ) Heidelberg, Faculty of Medicine and University Hospital Carl Gustav Carus, Technische Universität Dresden, Helmholtz-Zentrum Dresden-Rossendorf (HZDR), Dresden, Germany   [2]Medical Systems Biology, Medical Faculty Carl Gustav Carus, Technische Universität Dresden, Dresden, Germany   [3]Structural Bioinformatics, BIOTEC, Technische Universität Dresden, Dresden, Germany   [4]German Cancer Research Center (DKFZ), Heidelberg and German Cancer Consortium (DKTK) Partner Site, Dresden, Germany

Correspondence: mirko.theis@nct-dresden.de

is far from completion. In this study, we report a CRISPR/Cas9-based screen for the identification of genes eligible to increase NSCLC susceptibility to T cell–induced apoptosis when engaged by biAbs. The data provide functional insights into tumor escape mechanisms to lytic stress and present genes potentially suitable to increase the efficacy of anti-neoplastic therapy.

## Results

### Screen for pro-survival factors

To identify pro-survival factors of T cell–mediated cytotoxicity in NSCLCs, we performed pooled loss-of-function screening by CRISPR using Cas9 from *Streptococcus pyogenes* and a tailor-made sgRNA library targeting 1,572 human genes (Table S1) composed of 10,722 sgRNAs with a redundancy of 3–12 sgRNAs per gene. The library was delivered into the EGFR-mutated (L858R and T790M) NSCLC cell line NCI-H1975 (Cragg et al, 2007), which was chosen because of the significant cell surface expression of the tumor antigens erb-B2 receptor tyrosine kinase 2 (Her2), EGFR, and epithelial cell adhesion molecule (EpCAM) (Fig S1A), frequently targeted by biAbs in clinical trials (Runcie et al, 2018). After viral delivery, we first aimed at the identification of genes implicated in basic cellular processes such as cell growth or division. Accordingly, we determined changes in the sgRNA pool composition by deep sequencing after CRISPR/Cas9-mediated target gene knockout and subsequent cell culturing (Fig 1A). As controls, 632 sgRNAs targeting 45 viability and 47 non-essential genes (Evers et al, 2016) were included in the library (Table S1). We observed that 41 viability genes (91%) were depleted by at least fourfold on average (Fig 1B, Table S2), whereas sgRNAs targeting 38 non-essential genes (81%) were unaltered or mildly changed (average $\log_2 > -1$) (Fig 1B, Table S2), proving reliable control scoring. Beside these controls, we noticed that 218 genes were significantly depleted by at least fourfold on average and hence are putatively essential for cell growth or division (Fig 1B, Table S2). These genes were excluded from further analysis as we were aiming at factors involved in regulating NSCLC susceptibility to biAb/CTL-mediated killing. To reach this goal, we exposed tumor knockout cells to pre-activated human CD8+ T cells isolated from PBMCs and Her2-CD3 biAb in two subsequent rounds (Fig 1A) and determined library composition by deep sequencing at each time point. Furthermore, we challenged NCI-H1975 cells transduced with the sgRNA library with CTLs in the absence of biAbs to control for unspecific T cell effects. Strikingly, we observed that sgRNAs targeting the immune checkpoint gene *PD-L1* were largely unaltered at both time points (Fig 1C), in accordance with previous findings showing a minor relevance of *PD-L1* to CTL resistance in NSCLCs (Gainor et al, 2016). In contrast, sgRNAs targeting the *src family tyrosine kinase FYN* were depleted (Fig 1C), as expected from its role in cancer cell immune editing (Comba et al, 2020). Overall, these results corroborate the screening setup and its utility for the identification of genes impeding T cell–induced cytolysis of tumor cells.

To nominate candidate hit genes, we determined the sgRNAs depleted from the pool in the presence of biAbs in both consecutive

T cell treatments (T1: $\log_2 < -1$; T2: $\log_2 < -2$) (Fig 1A and C). To avoid genes involved in processes unrelated with T cell engagement, we excluded all sgRNAs depleted (C1: $\log_2 < -1$; C2: $\log_2 < -2$) (Fig S1B) in the control samples without biAbs (Table S2). Only genes with at least two independent sgRNAs scoring were considered. Finally, the gene *chromosome 22 ORF 46* (*C22orf46*) was chosen for further characterization as it showed robust scoring by two sgRNAs (Fig 1C and D) but no effects in the absence of biAbs (Fig S1B), indicating T cell–specific effects. Furthermore, we observed no significant effect of *C22orf46* depletion on cell viability (average across all sgRNAs $\log_2 = -0.73$) (Fig 1B) in the absence of T cells. In summary, the screen identified the gene *C22orf46* as a putative pro-survival factor in cancer cells subjected to cytotoxic effects of T lymphocytes, which is not essential for cell growth or division.

### *C22orf46* regulates T cell–induced apoptosis

To validate *C22orf46* as a pro-survival factor of T cell–induced cytotoxicity, we used CRISPR/Cas9 genome editing in NCI-H1975 cells to generate monoclonal knockout cell lines (Fig S2A). After confirming target gene knockout by four sgRNAs (Fig S2B) and depletion of the *C22orf46* transcript (Fig S2C), we measured apoptosis induction by biAb/CTL treatment using T cells from at least two different donors to exclude donor-specific effects. We detected a significant increase ($P < 0.05$) of apoptosis induction in all *C22orf46* knockout cell lines and in a *FYN* knockout compared with cells treated by a non-target (nt) control (Fig 2A and B). Furthermore, diminished cytotoxicity was observed in the absence of biAbs proving specific effects relying on T cell engagement (Fig S1C), thus verifying the results obtained in the screen (Fig 1). To determine the kinetics of T cell–mediated killing in cells devoid of *C22orf46* expression, we used a substrate of the executioner caspases 3 and 7 for detecting apoptotic cells and followed cell survival by fluorescence live cell image cytometry (Fig 2C). Strikingly, we observed an increase in the number of apoptotic cells after *C22orf46* knockout compared with a nt control starting several hours after addition of T cells and Her2-CD3 biAb (Fig 2C). After 10 h of exposure to biAb/CTLs, the rates of apoptosis were increased to 12% for cells devoid of *C22orf46* expression compared with the nt control (Fig 2C). No changes in tumor cell survival of knockout or control cells were detected when treated with CTLs in the absence of biAbs (Fig S1D).

To exclude that the observed pro-survival effects of *C22orf46* were limited to the cell line NCI-H1975 or to EGFR-mutated NSCLCs, we nominated the NSCLC cell line A549, which harbors unmutated EGFR alleles (Bai et al, 2017), and generated *C22orf46* knockout cell lines using the CRISPR/Cas9 technology. After verifying correct targeting of the corresponding genomic loci (Fig S2D) and transcript depletion (Fig S2D), we observed a significant increase in apoptosis induction by biAb/CTL treatment (Figs 2D and S1E) for four knockout cell lines in a range comparable with the effects observed in NCI-H1975 cells devoid of *C22orf46* expression (Fig 2A and B). We conclude that the effects of C22orf46 depletion are not limited to a particular NSCLC cell line or the presence of mutated EGFR.

Next, we investigated whether the sensitizing impact of the *C22orf46* knockout stemmed from the specific biAb used or its binding to the tumor antigen Her2. To examine this, we measured CTL cytotoxicity in the presence of a biAbs specific for EpCAM or

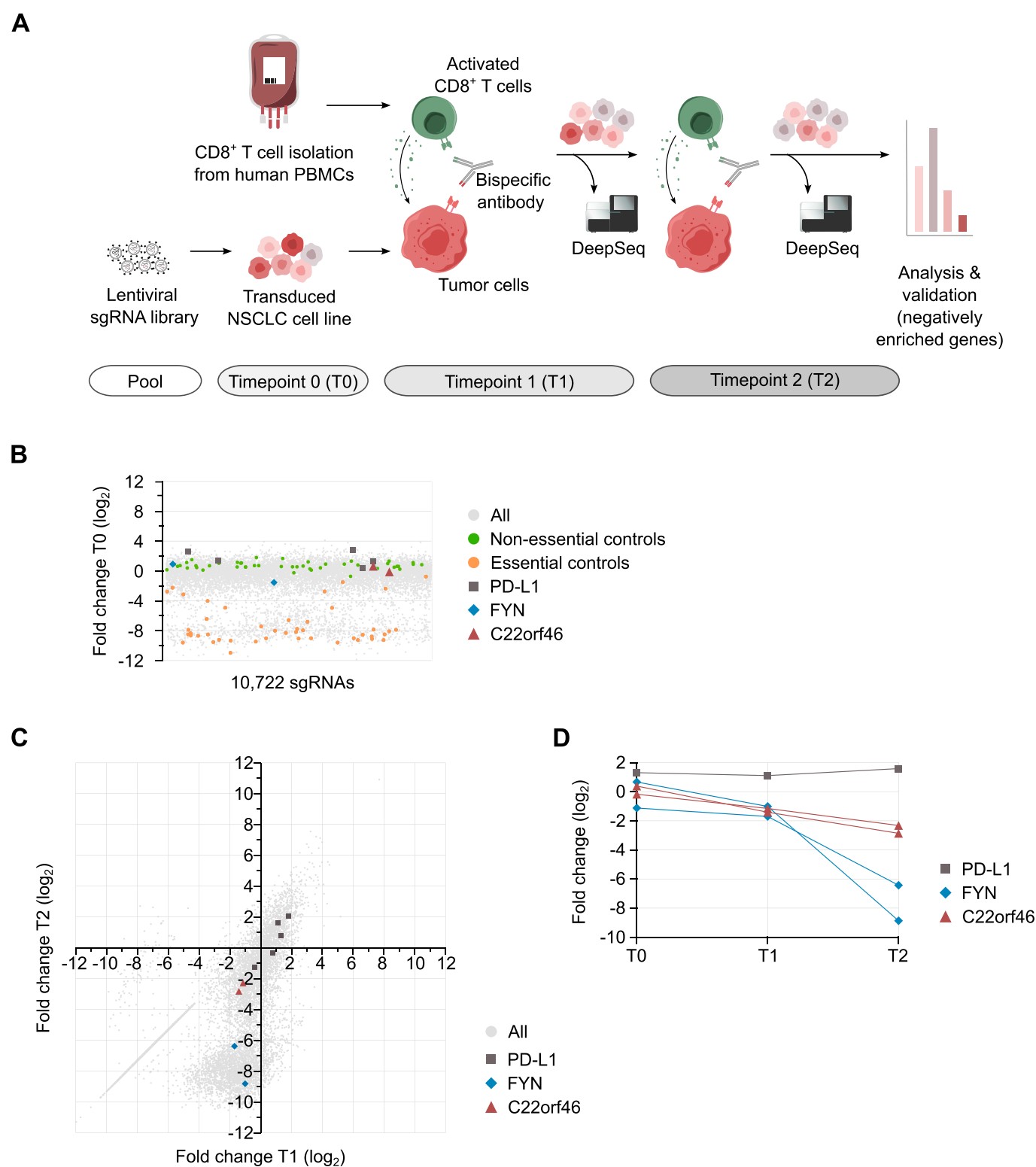

**Figure 1. CRISPR/Cas9 screen identifies pro-survival factors.**
**(A)** Schematic of the screen for pro-survival factors of T cell–mediated cytotoxicity in non-small cell lung cancers. NCI-H1975 cells were transduced with a sgRNA/Cas9 library. Knockout tumor cells (red) were targeted with pre-activated human CD8[+] T cells (green) engaged by a Her2-CD3 bispecific antibody in two subsequent rounds. Library composition was determined by deep sequencing at each time point. **(B)** Fold changes in the sgRNA pool composition (log$_2$) after gene knockout. sgRNA frequencies are shown (gray) and highlighted for *C22orf46* (red), the essential (orange) and non-essential (green) controls, *PD-L1* (black) and *FYN* (blue). **(C)** Fold changes in the sgRNA pool composition (log$_2$) after Her2-CD3 biAb/CD8[+] T cell targeting of NCI-H1975 knockout cells. Horizontal axis shows fold change (log$_2$) after first Her2-CD3 biAb/

EGFR and CD3. For both engagers, we detected an increase in apoptosis induction in *C22orf46* knockout cells compared to a nt control (Fig 2E) in the same range as expected from the Her2-CD3 biAb (Fig 2A and D), verifying effects independent of the cancer antigen.

BiAbs are often limited in their efficacy by inhibitory factors expressed by tumor cells such as immune checkpoint proteins (Junttila et al, 2014; Deisting et al, 2015). Consequently, we tested whether cells devoid of *C22orf46* expression show a concomitant down-regulation of the checkpoint proteins PD-L1 (Watson et al, 2020), CD47 (Tong & Wang, 2018), or B7-H3 (Yonesaka et al, 2018), explaining the phenotype. Interestingly, no significant changes in cell surface checkpoint expression were observed in *C22orf46* knockout cells compared with an nt control (Fig S1F, $MFI_{CP}/MFI_{nt}$ sgRNA1, sgRNA2, PD-L1: 1.25, 0.82; B7-H3: 1.23, 1.03; CD47: 1.20, 1.00). Typically, increased antigen expression leads to increased biAb efficacy. Hence, we hypothesized that *C22orf46* knockout may promote cell surface presentation of the tumor antigens Her2, EGFR, and EpCAM, providing an explanation for the observed phenotype. To investigate this possibility, we quantified biAb cell surface expression in *C22orf46* knockout and nt control cells and detected no significant alterations ($MFI_{AG}/MFI_{nt}$ sgRNA1, sgRNA2, Her2: 0.97, 0.68; EpCAM: 1.52, 0.94; EGFR: 1.03, 1.05) in the level of antigen presentation (Fig S1A). We conclude that the effects caused by *C22orf46* knockout were not attributed to changes in checkpoint or tumor antigen expression. Taken together, the data establish *C22orf46* as pro-survival factor of T cell–induced apoptosis in NSCLC independent of the EGFR mutational status, target tumor antigen and checkpoint expression.

### ORF *FLJ23584* of *C22orf46* encodes a nucleolar protein

*C22orf46* has been annotated as a non-protein coding gene, according to the NCBI GenBank (NR_160905.1). In contrast, two ORFs (C9J442, FLJ23584) have been reported at the Universal Protein Resource (UniProt) (UniProt Consortium et al, 2023). To characterize the gene structure of *C22orf46* and investigate the protein coding potential, we probed the splicing variants expressed in NCI-H1975 cells using various primers spanning exon-intron boundaries (Fig 3A). The data verified the expression of the transcript NCBI NR_160905 in NSCLCs (Fig 3A). In addition, we detected three alternative splicing variants, which were in accordance with transcripts reported by the Ensembl genome database (ENSG00000184208) with the transcript IDs ENST00000472110.3 and ENST00000668666.1 (Fig 3A). Interestingly, all transcript sequences were in accordance with the UniProt ORFs C9J442 and FLJ23584 that might encode for proteins with a molecular weight of 25 and 27 kD, respectively. To probe for protein expression, we endogenously inserted an eGFP reporter gene before the stop codon of FLJ23584 (Fig S3A) and verified genomic integration by PCR and Sanger sequencing (Fig S3B). Interestingly, eGFP expression was detected from this endogenous genomic insertion into the *FLJ23584* ORF (Fig 3B). Consequently, we

supposed that the ORF *FLJ23584* might encode for a translated protein. To corroborate this assumption, we determined the size of the expressed eGFP-tagged protein by Western blotting. Strikingly, we detected an eGFP signal at a molecular weight corresponding to eGFP-tagged *FLJ23584* protein (Fig 3C). To substantiate these data, we enriched the FLJ23584-eGFP protein by immunoprecipitation and determined the composition by mass spectrometry. Accordingly, we identified two peptides corresponding to the *FLJ23584* ORF (Figs 3D and S3C and D), verifying its identity. We conclude that the gene *C22orf46* is actively translated in NSCLCs, encoding a protein of 234 amino acids length on the ORF FLJ23584 (Fig 3D).

Notably, the ProteinPredict software (Bernhofer et al, 2021) predicted a nuclear localization of the protein C22orf46/FLJ23584. Furthermore, by manual inspection, two putative monopartite NLS (Lu et al, 2021) and a bipartite NLS (Lu et al, 2021) were identified (Fig 3D). To verify the nuclear localization of C22orf46/FLJ23584, we examined cells stably expressing *FLJ23584-eGFP* by immunofluorescence microscopy and detected a distinct nucleolar localization as deduced from a co-staining for the snoRNP methyltransferase fibrillarin (Reichow et al, 2007) (Fig 3E). To exclude effects caused by the eGFP tag, we used a vector encoding myc-DDK–tagged C22orf46/FLJ23584 and observed again a nucleolar localization (Fig S4A), corroborating the results obtained with the cells harboring an endogenous eGFP tag (Fig 3E). Taken together, the data show that the gene *C22orf46* encodes a nucleolar protein on the ORF *FLJ23584* expressed in NSCLCs.

### C22orf46/FLJ23584 exhibits structural features of the Bcl-2 family

To characterize the protein C22orf46/FLJ23584 functionally, we first searched for sequence homologies to already-characterized proteins. No significant sequence similarity was found to any protein or structure in the Protein Data Bank (PDB; https://www.rcsb.org). AlphaFold (Jumper et al, 2021a, 2021b) predictions showed a mainly disordered protein with the exception of three α-helices predicted with "*very high*," "*confident*," and "*low/very low*" confidence level, respectively (https://alphafold.ebi.ac.uk/entry/Q9H5C5). Unfortunately, the predicted three-dimensional disposition of these three helices was not enough to structurally assign a 3D fold. Secondary structure predictions with PSIPRED confirmed those three helices and other two with high level of confidence (Fig S5A). We therefore decided to use threading-based fold recognition methods (i.e., Phyre² [Kelley et al, 2015] and I-Tasser [Yang et al, 2015]) to investigate the 3D fold that C22orf46/FLJ23584 could possibly adopt. Although a substantial part of the protein was predicted to be disordered by threading (i.e., 54%), we identified a segment of 20 residues of the sequence predicted to resemble with high confidence helical segments of the structure of the BH3-interacting domain death agonist (Bid) (i.e., 71% confidence level in Phyre² with the NMR structure of mouse Bid; PDBID 1DDB [McDonnell et al, 1999] [Fig S5B] and within the top 10 templates matched with I-Tasser [Fig S5C]). Bid is an apoptotic regulator and member of the Bcl-2 family. Bcl-2 has

---

CD8[+] T cell treatment (T1) and vertical axis represents fold change (log$_2$) after second treatment (T2) in gray. sgRNAs targeting *PD-L1* (black), *FYN* (blue) and *C22orf46* (red) are highlighted. **(D)** Fold changes in sgRNA frequencies (log$_2$) for T0, T1, and T2 for *PD-L1* (black, one representative sgRNA shown), *FYN* (blue) and *C22orf46* (red).

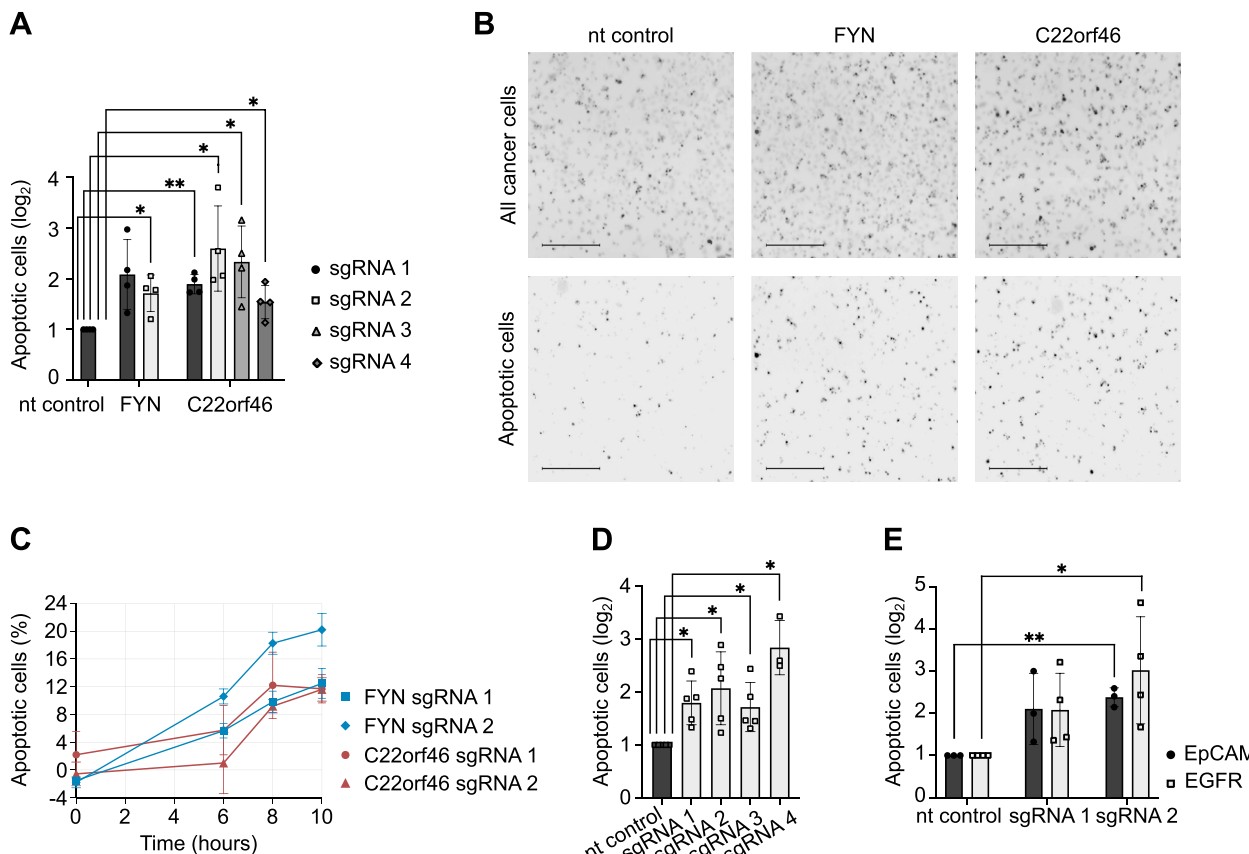

**Figure 2.** *C22orf46* knockout enhances T cell–induced apoptosis.
**(A)** Relative numbers of apoptotic NCI-H1975 cells for nt control, *FYN* or *C22orf46* knockout with different sgRNAs treated with Her2-CD3 biAb and CD8[+] T cells from two donors (log$_2$ of mean fold change normalized to nt control, ±SD; n = 4 biological replicates; Welch's two-tailed *t*-test, *$P$ < 0.05, **$P$ < 0.01). **(B)** Representative images from fluorescence live cell image cytometry of nt control (left), *FYN* (middle) and *C22orf46* knockout (right) NCI-H1975 cells treated with Her2-CD3 biAb and CD8[+] T cells. All cancer cells were stained with CellTracker Orange CMRA dye (upper row) and apoptotic cells were monitored via Incucyte Caspase-3/7 Green dye (lower row) (scale bars = 500 μm). **(C)** Time course measurements for *FYN* (blue) and *C22orf46* (red) knockout NCI-H1975 cells treated with Her2-CD3 biAb/CD8[+] T cells. Percentages of apoptotic cells for a nt control (not shown) were subtracted (n = 3, technical replicates). **(D)** Relative numbers of apoptotic nt control (black) or *C22orf46* knockout (gray) A549 cells treated with Her2-CD3 biAb/CD8[+] T cells (log$_2$ of mean fold change normalized to nt control, ±SD; n = 5/3 biological replicates, three T cell donors; Welch's two-tailed *t*-test, *$P$ < 0.05). **(E)** Relative numbers of apoptotic nt control or *C22orf46* knockout NCI-H1975 cells (log$_2$ of mean fold change normalized to nt control, ±SD; n = 3/4 biological replicates, three T cell donors) treated with EpCAM-CD3 (black) or epidermal growth factor receptor-CD3 (gray) biAb/CD8[+] T cells (Welch's two-tailed *t*-test, *$P$ < 0.05, **$P$ < 0.01).

been reported to facilitate anti-apoptotic functions with relevance in tumorigenesis (Kaloni et al, 2023). Proteins belonging to the Bcl-2 family contain Bcl-2 homology (BH) domains constituted by a short helical fragment, which exhibit low sequence similarity within the family (McDonnell et al, 1999). BH domains contribute at multiple levels to the function of these proteins in cell death and survival. These short helical BH domains interact among each other inter- and intra-molecularly, conferring the proteins apoptotic regulatory properties (Chou et al, 1999). Taking into account the obtained secondary structure predictions and threading results, we explored in more detail the possible resemblance of the predicted helical regions of C22orf46/FLJ23584 to BH domains of the Bcl-2 protein family. A structure-based sequence alignment of C22orf46/FLJ23584 to Bid and a representative set of other members of the Bcl-2 family shows that several regions of the protein indeed exhibit resemblance to BH domains (Fig 4A). Based on these observations, we propose that C22orf46/FLJ23584 contains three putative BH domains, which

implies that it might be a new member of the Bcl-2 protein family (McDonnell et al, 1999).

### *C22orf46/FLJ23584* impedes pro-apoptotic signaling

The gene *C22orf46* encodes a nucleolar protein (Fig 3E), which diminishes T cell–mediated cytotoxicity in NSCLC cells (Fig 2) and has been reported to be significantly up-regulated in a broad range of various cancer entities including lung carcinomas ($P$ = 2.5 × 10$^{-30}$) (Bartha & Gyorffy, 2021) (Fig 4B). Furthermore, C22orf46/FLJ23584 shows structural features of Bcl-2 family members (Fig 4A) known to facilitate anti-apoptotic functions with relevance in tumorigenesis (Kaloni et al, 2023). These data prompted the investigation of whether the sensitizing effects seen by CTLs on *C22orf46* knockout cells could be because of a general anti-apoptotic function. To test this assumption, we treated cells devoid of *C22orf46* expression with the genotoxic substance camptothecin (Li et al, 2006) and measured apoptosis by determining caspase 3/7 activity.

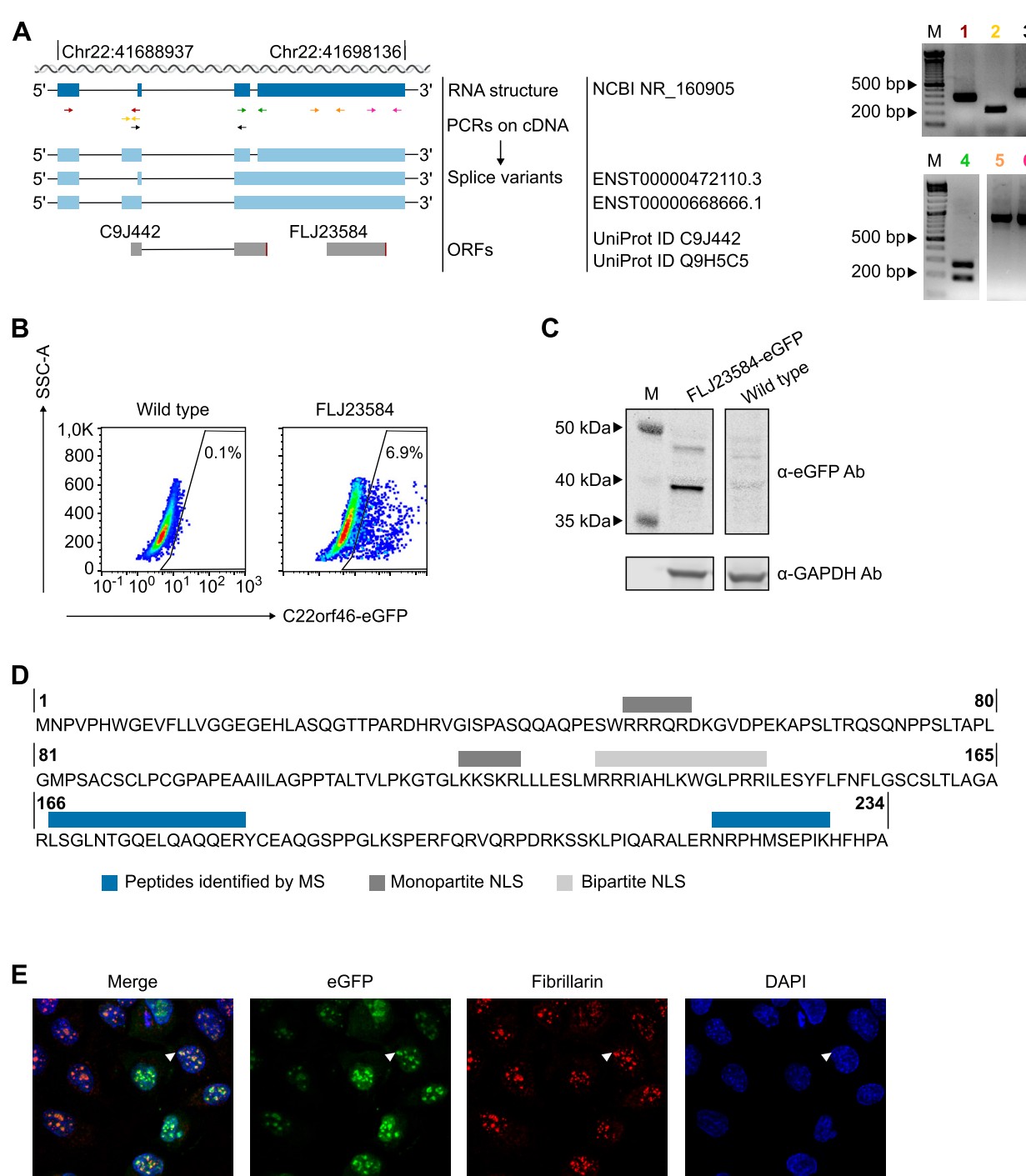

**Figure 3. *C22orf46/FLJ23584* encodes a nucleolar protein.**
**(A)** Schematic (left) of *C22orf46* transcript structures (chromosomal (chr) position and ORFs as indicated). The *C22orf46* transcript NR_160905 (dark blue) shown with primer positions (arrows, different colors). Alternative splice variants (light blue) with Ensembl IDs and ORFs (gray) annotated by UniProt as depicted. Agarose gels (right) of PCR products with the indicated primer pairs (1–6, different colors) from NCI-H1975 cDNA. Marker band sizes (M) as indicated. **(B)** Flow cytometry plots of eGFP⁺ fluorescence and side scatter of NCI-H1975 WT cells (left) and eGFP-tagged *C22orf46* ORF *FLJ23584* (right) cells. Percentages of cells in the eGFP⁺ gates are indicated. **(C)** Total cellular protein levels of eGFP (top) and GAPDH (bottom) in NCI-H1975 WT (right) and *FLJ23584*-eGFP-tagged cells (left) visualized by immunoblotting. Marker band sizes (M) as indicated. **(D)** Schematic of the amino acid sequence for the *C22orf46* ORF *FLJ23584*. Bars indicating sequences of peptides identified by mass spectrometry (MS, blue) and monopartite (dark gray) and bipartite (light gray) NLS. **(E)** Immunofluorescence microscopy images of *FLJ23584*-eGFP-tagged NCI-H1975 cells stained with a fibrillarin antibody and DAPI. Arrowheads indicate *FLJ23584*-eGFP protein localized to nucleolar structures (scale bar: 25 µm).

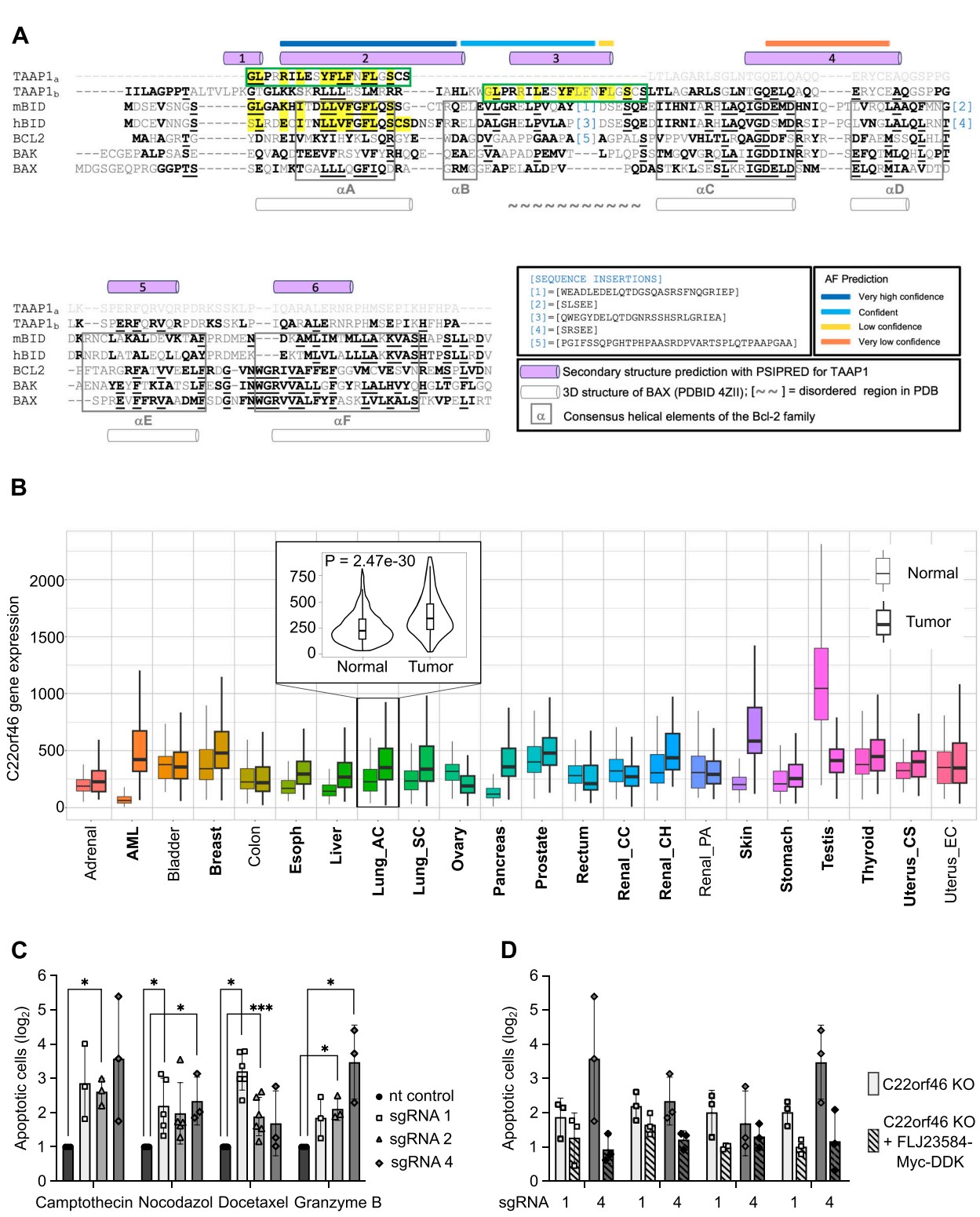

**Figure 4. Structure-based sequence alignment of C22orf46/FLJ23584 and pro-survival functions.**
**(A)** Structure-based sequence alignment of C22orf46/FLJ23584 with a set of representative members of the Bcl-2 protein family. Predictions obtained with AlphaFold (AF) for C22orf46/FLJ23584 are highlighted at the top of the alignment in AF-confidence color code. The secondary structure α-helical prediction for C22orf46/FLJ23584 obtained with high confidence from PSIPRED is depicted by violet cylinders at the top of the alignment and numbered (helix 1–6). Boxed regions represent the consensus helical elements of the Bcl-2 family. The 20-residue long segment of C22orf46/FLJ23584 identified by threading to resemble a BH domain in BID is shown in a green box at the top (TAAP1a) with sequences matching highlighted in yellow. This segment corresponds to helix-3 in the full alignment (TAAP1b), which is shown below (green box).

Camptothecin inhibits topoisomerase I (Martin-Encinas et al, 2022) but has also been demonstrated to disrupt transcription of RNA polymerase I (Pol I) (Garg et al, 1987; Pondarre et al, 1997) and might hence interfere with functions of the nucleolus, which has been shown to depend on Pol I activity (Quin et al, 2014). Strikingly, we observed a significant increase in NSCLC susceptibility to apoptosis induction by camptothecin in *C22orf46* knockout cells compared with a nt control (Fig 4C). To test whether this effect could be attributed to topoisomerase I or Pol I inhibition, we examined the spindle toxins nocodazole (Beswick et al, 2006) and docetaxel (Montero et al, 2005) for their effects on *C22orf46* knockout cells. Interestingly, we detected sensitizing effects of the loss of C22orf46/FLJ23584 expression to nocodazole- or docetaxel-induced apoptosis as observed for camptothecin (Fig 4C). We conclude that C22orf46/FLJ23584 might have pro-survival functions, sensitizing cancer cells for apoptosis when impaired.

CTLs partially mediate their cytotoxic effects on tumor cells by secreting the serine protease granzyme B (GZMB), which cleaves and activates the initiator caspases 8 and 10 (Afonina et al, 2010) and the pro-apoptotic protein Bid to trigger apoptosis (Kaloni et al, 2023). Hence, we asked if the sensitizing effects of C22orf46/FLJ23584 depletion for T cell–mediated apoptosis (Fig 2) depend on the presence of live T cells. To test that, we applied GZMB to *C22orf46* knockout cells and measured apoptosis by determining caspase 3/7 activity. Interestingly, we observed an increase in susceptibility of *C22orf46*-deficient cells to GZMB-induced apoptosis as compared with the anti-neoplastic drugs (Fig 4C) and as expected from the experiments performed with biAb/CTLs (Fig 2A). To corroborate these data, we tested whether the knockout phenotype can be reversed by the expression of *C22orf46* and transfected NCI-H1975 knockout cells with a vector encoding FLJ23584-myc-DDK. After proving exogenous expression (Fig S4B), we measured apoptosis induction and observed a rescue of the *C22orf46* knockout phenotype (Fig 4D).

In summary, the data promote a general anti-apoptotic function of nucleolar *C22orf46*/*FLJ23584* in NSCLCs. Thus, we suggest naming the protein encoded by the *ORF FLJ23584* of the gene *C22orf46* "Tumor Apoptosis Associated Protein 1 (TAAP1)."

## TAAP1/*C22orf46* modulates cancer-associated genes

Recent advances have revealed a compelling function of the nucleolus in regulating stress responses (Quin et al, 2014) beyond its canonical role in ribosome biogenesis (Quin et al, 2014). In addition, the nucleolus has been implicated into signaling events altering the activity of various transcription factors (Quin et al, 2014; Lindstrom et al, 2018). Hence, we assumed that cells with impaired

TAAP1 function might show changes in their transcriptional landscape. To test this hypothesis, we determined the transcriptome composition in *C22orf46* knockout cells and observed a concomitant depletion of TAAP1 transcript along with the up- ($\log_2 > 1$, $\log_{10} P < 1.5$) or down-regulation ($\log_2 < -1$, $\log_{10} P < 1.5$) of 543 and 59 genes, respectively (Fig 5A). Amongst the up-regulated genes, we identified factors known to foster anti-tumor immunity such as CXCL11 ($\log_2 = 2.2$) (Tokunaga et al, 2018), IL-10 ($\log_2 = 2.3$) (Oft, 2014), and ICAM-1 ($\log_2 = 1.3$) (Reina & Espel, 2017) (Fig 5A).

To get a more comprehensive analysis of the altered gene functions, we performed gene set enrichment analysis (GSEA) (Subramanian et al, 2005) using the Hallmark (Liberzon et al, 2015), Biocarta (Nishimura, 2004), and Kyoto Encyclopedia of Genes and Genomes (Kanehisa et al, 2023) collections obtained from the Molecular Signatures Database (http://www.broad.mit.edu/gsea/). Interestingly, we observed 28 significantly ($P < 0.01$) enriched gene sets (Fig S6) of which 12 were associated with immunological processes such as "cytokines and inflammatory response" (ES: 0.76, Biocarta), "inflammatory response" (ES: 0.54, Hallmark), and "TNF$\alpha$ signaling via NF$\kappa$B" (ES: 0.50, Hallmark) (Figs 5B and S6), implicating a role of *C22orf46* in shaping tumor expression profiles towards a more carcinogenic state involving immune-regulatory genes.

EGFR-targeted tyrosine kinase inhibitors have become the first-in-line treatment for EGFR-driven NSCLCs because of promising initial response rates and extended progression-free survival (Selenz et al, 2022). Remarkably, we have seen a significant (ES = −0.59, $P < 0.01$) overlap of the gene set down-regulated in EGFR inhibitor-treated NSCLCs (Kobayashi et al, 2006) and the genes altered by *C22orf46* knockout (Fig 5B) when using the "chemical and genetic perturbations" (CPG, customized for lung cancer) collection. The data indicate that targeting *C22orf46* in tumors might provide beneficial effects analogous to the effects observed by EGFR inhibition. Along these lines, we observed that *C22orf46* knockout led to the significant depletion of genes (ES = −0.4, $P < 0.01$) previously shown to correlate with poor survival in NSCLCs (Fig 5B) (Director's Challenge Consortium for the Molecular Classification of Lung Adenocarcinoma et al, 2008), implicating that *C22orf46* might contribute to aggressiveness of lung carcinomas. To corroborate this assumption, we analyzed the survival probability of 253 lung adenocarcinoma patients and observed a significant correlation ($P = 0.03$) of *C22orf46* expression and prognosis using data from the Cancer Genome Atlas (TCGA) (Fig 5C). In addition, we analyzed the frequency of somatic mutations (SNP or INDELs) in *C22orf46* by UCSC Xena (Goldman et al, 2020) and observed a low mutation rate amongst 514 primary tumor samples according to the TCGA lung adenocarcinoma (LUAD) data set, as expected from an anti-apoptotic oncogene (Delbridge et al, 2016).

The α-helices of the 3D structure of BAX (PDBID 4ZII) are depicted by white cylinders at the bottom of the alignment. For clarity, large sequence insertions have been marked with numbers and omitted from the alignment (shown under sequence insertions). mBID and hBID: mouse and human BH3-interacting domain death agonist, respectively. BCL-2: human apoptotic regulator Bcl-2. BAX: human apoptotic regulator BAX. BAK: human Bcl-2 homologous antagonist/killer. Alignment based on McDonnell et al (1999) and adapted to C22orf46/FLJ23584. **(B)** Expression range of *C22orf46* in normal and tumor tissue. RNA sequencing data are displayed as boxplots. Significant differences are marked in bold letters. Smaller graph displays detailed analysis of *C22orf46* expression in lung adenocarcinoma as violinplot. *P*-values determined with Mann-Whitney *U* test. Data and analysis obtained from TNMplot.com. **(C)** Relative numbers of apoptotic nt control (black) or *C22orf46* knockout (gray) NCI-H1975 cells ($\log_2$ of mean fold change normalized to nt control, ±SD; n = 2/3/5 biological replicates) treated with camptothecin, nocodazol, docetaxel or granzyme B as indicated (Welch's two-tailed *t*-test, *$P < 0.05$, ***$P < 0.001$). **(D)** Relative numbers of apoptotic *C22orf46* knockout NCI-H1975 cells and *C22orf46* knockout cells transfected with a FLJ23584-myc-DDK encoding vector ($\log_2$ of mean fold change normalized to the respective nt control ± SD; n = 2/3/5 biological replicates) treated as indicated.

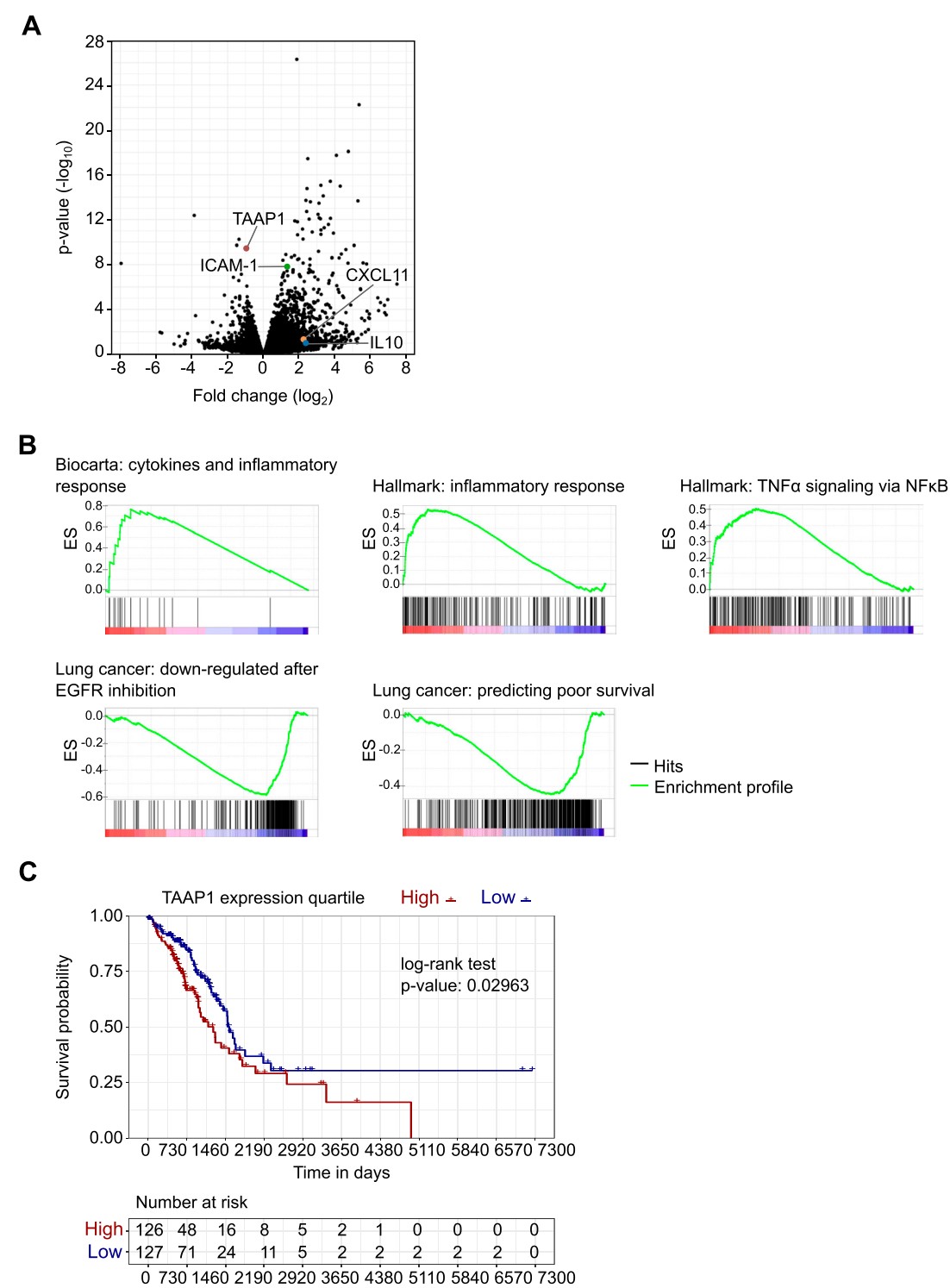

**Figure 5.  Transcriptome analytics reveal pro-tumorigenic function of TAAP1/*C22orf46*.**
**(A)** Volcano plot of a pairwise comparison of transcript expression in polyclonal TAAP1/*C22orf46* knockout NCI-H1975 cells compared with nt control. Horizontal axis shows $\log_2$ fold change in transcript expression and vertical axis represents statistical significance (negative $\log_{10}$ of $P$-value, n = 3). Transcripts of genes involved in anti-tumor immunity and TAAP1/C22orf46 are highlighted. **(B)** Enrichment plots obtained by gene set enrichment analysis of pre-ranked RNA-seq data from polyclonal *C22orf46* knockout NCI-H1975 cells compared with nt control. Significantly ($P$ < 0.01) enriched or depleted gene sets obtained from Biocarta, Hallmark or the "chemical and genetic perturbations" (CPG, customized for lung cancer) collections as part of the Human Molecular Signatures Database are shown. Top portion of the plot shows the running enrichment score (ES). Down portion of the plot shows the position in the ranked list of total genes for all members of the gene set. **(C)** Estimate of survival curves using the Kaplan-Meier method and the corresponding risk table for two groups of patients diagnosed with lung adenocarcinoma from the Cancer Genome Atlas Pan-

We conclude that impaired TAAP1 function in NSCLCs facilitates gene expression changes fostering anti-tumor immune responses associated with a better prognosis, along with alterations increasing the efficacy of anti-neoplastic drugs. Taken together, TAAP1/*C22orf46* is a potential target gene to increase cancer vulnerability to T cell–mediated cytotoxicity and chemotherapeutics by reducing pro-survival signaling.

## Discussion

NSCLC is the most frequent cancer type globally (Bray et al, 2018). Although innovation has led to remarkable improvements in therapy, the prognosis for NSCLC patients remains dismal (Bray et al, 2018). Accumulating lines of evidence indicate that aberrant expression or dysregulation of anti-apoptotic proteins play a decisive role in NSCLCs by mediating resistance to the immune system (Tuomela et al, 2022) or to anti-neoplastic drugs (Strasser et al, 1994; Kaloni et al, 2023), contributing significantly to cancer progression (Delbridge et al, 2016) and metastasis (Choi et al, 2016). For example, the oncogene Bcl-2 has been reported to promote tumorigenesis by enabling cell survival (Delbridge et al, 2016). Bcl-2 gave rise to a growing family of at least 16 proteins that bear one or more Bcl-2 homology (BH) domains and are implicated in facilitating regulatory functions in cell destiny decisions (Delbridge et al, 2016). BH3-only proteins belong to the Bcl-2 family and have been reported to work as sentinels of intracellular damage (Danial & Korsmeyer, 2004). For example, the BH3-containing protein Bid has been implicated in regulating apoptosis following DNA damage (Kamer et al, 2005) and localizes to the nucleolus (Kamer et al, 2005; Zinkel et al, 2005), where it is a target of the kinase ATM, which is critical for cell cycle arrest and apoptosis induction (Zinkel et al, 2005). Several Bcl-2 family proteins are frequently overexpressed in various malignancies including lung cancer (Ikegaki et al, 1994), and tumor cells can be sensitized for anti-neoplastic agents by treatment with inhibitors of Bcl-2 family proteins (Oltersdorf et al, 2005; Cragg et al, 2007).

We identified the gene *C22orf46* to conduct pro-survival signaling in lung adenocarcinoma and predicted that TAAP1 encoded by *C22orf46* contains regions resembling BH domains, implying that TAAP1 may hence be a putative member of the Bcl-2 protein family of apoptosis regulators. Accordingly, we observed that downregulation of the gene *C22orf46* sensitizes cancer cells for T cell–mediated lysis and for apoptosis induced by chemotherapeutic agents. Furthermore, *C22orf46* expression correlates with adverse prognosis in lung adenocarcinomas and is significantly up-regulated in various cancer entities. In this context, it is remarkable that the circular RNA circ-CCAC1 has been reported to be overexpressed in adrenocortical carcinoma, and its expression correlates with poor prognosis (Li et al, 2020). circ-CCAC1 enhances *C22orf46* expression by sponging miR-514a-5p, a microRNA with tumor suppressor activity (Xiao et al, 2018), binding the 3′-UTR of the

*C22orf46* transcript (Li et al, 2020). In addition, another study reported *C22orf46* as a biomarker with a significant involvement in the pathogenesis of a subtype of adult T cell leukemia (Zarei Ghobadi et al, 2023). Hence, we speculate that the oncogenic effects reported for circ-CCAC1 and miR-514a-5p are at least partially mediated by TAAP1/*C22orf46* in various cancer types.

It has been reported that anti-apoptotic proteins are regulated by diverse mechanisms and seemingly at cellular localizations in which they can sense or communicate specific damage (Popgeorgiev et al, 2018). For instance, the intracellular distribution of Bcl-2 proteins is dynamic and profoundly affected by stress signaling (Popgeorgiev et al, 2018). Interestingly, Bcl-2 has been reported to localize to the nucleus in various cancer types including breast cancer, endometrial carcinoma, squamous cell carcinoma, and astrocytoma (Chan et al, 1995; Mosnier et al, 1996; Choi et al, 2016), where it appears to take part in a multiprotein complex comprising CDK1, PP1, and nucleolin (Barboule et al, 2005, 2009). Whereas the primary function of the nucleolus is related to ribosome production (Quin et al, 2014), it is increasingly recognized as having an additional role in the regulation of cell fate decisions and gene expression with profound implications to carcinogenesis (Hua et al, 2022). This assumption is supported by the characteristics of cancer cells to display large and increased numbers of nucleoli (Derenzini et al, 2009), and enlargement of nucleolar size correlates with adverse prognosis (Ruggero, 2012; Samaratunga et al, 2014). Furthermore, changes in nucleolar size have become a well-established histopathological parameter for assessing the response to chemotherapeutics (Derenzini et al, 2009). Noteworthy, of over 4,500 proteins reported to localize to the nucleolus less than half have defined functions in ribosome biogenesis further promoting the importance of the nucleolus in processes other than ribosome assembly (Quin et al, 2014). Rather, they are involved in a diverse range of functions including tumor suppressor and proto–oncogene activities, regulating cell cycle control or stress signaling (Quin et al, 2014). Hence, it is not surprising that dysregulation of ribosome biogenesis and nucleolar structure has been associated with an increased risk of cancer (Quin et al, 2014). Along these lines, we observed that the protein TAAP1 localizes to the nucleolus in NSCLCs. Consequently, we hypothesize that TAAP1 might be a sentinel of nucleolar stress, contributing in pro-survival signaling. Accordingly, we observed that expression of TAAP1 leads to transcriptome alterations correlating with worse prognosis in lung cancer and conversely that knockout of *C22orf46* results in beneficial transcriptional changes associated with a better prognosis. Moreover, we discovered alterations in the expression of immune-regulatory genes in cells devoid of *C22orf46* expression, enhancing tumor visibility to the immune system such as CXCL11, ICAM-1, or IL-10. The chemokine C-X-C motif chemokine ligand 11 (CXLC11) has been implicated to increase the frequency of tumor-infiltrating lymphocytes and inhibits tumor growth in colon cancer (Cao et al, 2021). Furthermore, CXCL11 has been established as an independent prognostic biomarker promoting anti-tumor immunity and prolonged overall survival in lung carcinoma patients (Gao

---

Cancer atlas. The groups labeled as "high" and "low" are subsets of patients with *C22orf46* gene expression values from tumor samples above the upper or below the lower quartile, respectively. The *P*-value was computed using a log-rank test.

et al, 2019). Expression of *intercellular adhesion molecule 1* (*ICAM-1*) has been correlated with a better prognosis in colorectal (Wimmenauer et al, 1997; Maeda et al, 2002) and lung cancer (Haustein et al, 2014). In addition, mouse melanoma tumors that relapse after adoptive T cell therapy show decreased content of ICAM-1 mRNA (Straetemans et al, 2015). The cytokine IL-10 has been implicated in anti-tumor effects as humans deficient of IL-10 develop tumors spontaneously and at high rates (Oft, 2014). Furthermore, IL-10 has been shown to stimulate cytotoxicity of CD8[+] (Oft, 2014) and CAR T cells (Gorby et al, 2020), IFNγ expression, and up-regulation of MHC molecules (Rallis et al, 2021). Consequently, PEG-IL-10 has been explored as therapeutic agent concomitantly stimulating T cell responses and encouraging chronic versus acute inflammation both shown to foster anti-tumor immunity (Rallis et al, 2021). We conclude that the protein TAAP1 contributes to pro-tumorigenic nucleolar signaling and gene expression, affecting the interaction of tumor and host immune cells by providing pro-survival signals under stress conditions.

However, the functional data on TAAP1 are limited to conclusions drawn by in vitro experiments. Although patient data show a correlation of *C22orf46* expression with prognosis, a study on *C22orf46* in a cellular system closer to the physiological situation or in a model organism is required to verify the assumptions drawn. Importantly, nucleoli are dynamic in nature and exquisitely regulated by multiple signaling pathways (Quin et al, 2014). It would be very interesting to investigate whether *C22orf46* is actively involved in these dynamic processes and to which extend they depend on extracellular stress by genotoxic substances or immune cells. Significantly, these pathways inherently contain oncogenes and tumor suppressors, and their dysregulation enables cells to achieve the uncontrolled growth and proliferation that is a hallmark of cancer (Hanahan, 2022).

The growing evidence that implicates the nucleolus in vital processes of carcinogenesis supports the idea of a functional contribution of nucleolar proteins such as TAAP1. However, central components of the apoptosis machinery reside in the cytoplasm, and the interaction partners of TAAP1 in the nucleolus that allow signal transduction remain unidentified. Further research in this area is required to unravel the intricate relationship between the nucleolus and pro-survival oncogenes. To sum up, the nucleolar protein TAAP1 encoded by *C22orf46* comprises anti-apoptotic functions in NSCLC, and its inhibition reduces tumor-associated immune suppression and pro-survival signaling, fostering the efficacy of anti-tumor therapies.

# Materials and Methods

## Cell culture

The NSCLC cell lines NCI-H1975 (CRL-5908) and A549 (CCL-185) were purchased from ATCC and maintained in RPMI medium (21875; Gibco) supplemented with 10% FBS (A5256701; Gibco), 100 units/ml penicillin, and 0.1 mg/ml streptomycin (P06-07100; Pan Biotech) at 37°C and 5% $CO_2$. HEK293T (CRL-11268) cells were cultured in DMEM medium (31966; Gibco) as described above. To ensure optimal experimental conditions, a recovery period of two passages was provided for all cell lines after thawing. Passaging of cells took place when they reached a confluency of 70–80%. EDTA-treated whole blood samples were collected from healthy donors, and PBMCs were isolated using Lymphoprep (1114545; Stemcell Technologies) through density gradient centrifugation. Subsequently, CD8[+] T cells were isolated from PBMCs via negative selection using the CD8[+] T cell isolation kit (130-096-495; MACS, Miltenyi Biotec), followed by expansion and activation using Dynabeads human T-Activator CD3/CD28 (11131D; Gibco) in ImmunoCult-XF T cell expansion medium (10981; Stemcell Technologies). After 72 h, activation beads were removed, and CD8[+] T cells were maintained in culture for an additional 24 h before conducting the functional tests.

## CRISPR library

A sgRNA library was designed to cover six protein classes including kinases, nuclear receptors, cell surface proteins (Bausch-Fluck et al, 2015), epigenetic factors, transcription factors, and uncharacterized genes. Each gene was targeted with 3–12 different sgRNAs, selected to specifically bind to either the first exon, an early splicing site, or the functional domain of the protein. All sgRNAs were chosen to fulfill the criteria defined by Doench et al (2016). The complete library consisted of 10,722 sgRNAs targeting 1,572 genes. 632 sgRNAs were included to target 45 essential and 47 non-essential control genes. Oligonucleotides with sgRNA sequences were ordered as arrayed synthesis from CustomArray Inc. and PCR amplified (primer, forward: 5'-GATATTGCAACGTCTCACACC-3', reverse: 5'-GTCGCGTACGTCTCGAAAC-3'). The resulting PCR product was cloned into the lentiviral vector pL.CRISPR.EFS.tRFP (57819; Addgene) using the Esp3I restriction site. The vector contained a modified tracr sequence (5'-GTTTAAGAGCTATGCTGGAAACAGCATAGCAAGTTTAAA-TAAGGCTAGTCCGTTATCAACTTGAAAAAGTGGCACCGAGTCGGTGCTTTTTTT-3') as previously described (Chen et al, 2013).

## Lentivirus production and CRISPR screen

To produce lentivirus, HEK293T cells were transfected with 10 μg of the vector pL.CRISPR.EFS.tRFP (57818; Addgene) encoding for *S. pyogenes* Cas9 and a sgRNA from the library, respectively. In addition, 6 μg psPAX2 (12260; Addgene) and 2 μg pMD2.G (12259; Addgene) packaging plasmids as well as 45 μg polyethylenimine (408719; Sigma-Aldrich) were used for the transfection process. After 72 h, the virus-containing supernatant was collected, concentrated through centrifugation using Amicon Ultra-15 centrifugal filters (UFC810024; Merck) according to manufacturer's instructions and diluted with cell culture medium at a ratio of 1:2. For long-term storage, the virus particles were preserved in cryovials at −80°C. A titration was performed to determine the virus quantity necessary to infect 30% of the cells, corresponding to a multiplicity of infection (MOI) of 0.5. Then, the appropriate amounts of virus were added to NCI-H1975 WT cells, followed by centrifugation at 1,000g for 30 min, a 24 h incubation period, and medium change. 8 d after infection, the cells were subjected to FACS using a BD FACSAriaIII cell sorter. A total of 3.7 million tRFP-positive cells were collected.

After sorting, an additional 8-d recovery period (16 d post-transduction) was allowed before the cells were split into two populations. Genomic DNA extraction was performed on five million cells using the QiAmp DNA blood kit (51104; QIAGEN) as described in the manufacturer's protocol. The remaining population was maintained in culture, and at 22 d post-transduction, 2.2 million cells were seeded on a 100 mm dish. After 24 h, the cancer cells were exposed to T cells at a ratio E:T = 2:1 with 0.25 µg/ml Her2/CD3 bispecific antibody. After an incubation of 8 h, NCI-H1975 cells were detached by trypsin, and T cells were separated using the EasySep Human CD8 Positive Selection Kit II (Stemcell) following the manufacturer's protocol. The cancer cells were cultured for an additional 7 d, after which one portion of the population was used for genomic DNA extraction, whereas the other portion was prepared for a second treatment with T cells as described above.

Isolated gDNA samples from sorted cells 15 d post-transduction and after round one and two of T cell treatment were used to amplify sgRNA sequences by two rounds of PCR, with the second-round primers containing adaptors for Illumina sequencing (PCR 1, forward 5′-GTAATAATTTCTTGGGTAGTTTGCA-3′, reverse 5′-ATTGTGGATGAATACTGCCATTTG-3′; PCR 2, forward 5′-ACACTCTTTCCC-TACACGACGCTCTTCCGATCTGGCTTTATATATCTTGTGGAAAGG-3′, reverse 5′- GTGACTGGAGTTCAGACGTGTGCTCTTCCGATCTCAAGTTGATAACGGACT AGCC-3′). The resulting libraries were sequenced with single-end reads on a NextSeq 500. In brief, after targeted PCR amplification, the samples were indexed for NGS sequencing in a successive PCR enrichment followed by purification and capillary electrophoresis (Fragment Analyzer; Agilent). The sequence reads were mapped to sgRNA sequences with the aid of PatMaN (Prufer et al, 2008), a rapid short sequence aligner. As a set of query patterns, we used sgRNA sequences flanked by 5′-GACGAAACACCG-3′ and 5′-GTTTAAGAGCTA-3′ on the termini, respectively, and allowed two mismatches during the alignment step. For each read, the best matching gRNA sequence was picked, and in case of ties, the read was discarded as ambiguous. Finally, for each sequenced sample, counts of reads mapped to each sgRNA from the library were calculated.

## CRISPR/Cas9 knockout cell lines

For targeted gene knockout, sgRNAs were designed to target the respective gene of interest. As control, a sgRNA with no homologies to the human genome was used (non-target control). Sequences are given in Table S3. Transfection of 0.2 million NCI-H1975 or A549 cells with 0.5 pmol sgRNA, 1.3 µg CleanCap Cas9 mRNA (L-7606; TriLink), and 0.2 µl eGFP mRNA was performed using the 4D-NucleofectorTM X Unit (Lonza). The SF Cell Line 4D-Nucleofector X Kit S (V4XC-2032; Lonza) was used following the manufacturer's protocol. The transfection efficiency was assessed by measuring eGFP expression 1 d after transfection and has typically exceeded 90%. After a culture period of 2 wk, genomic DNA (gDNA) was isolated from the transfected cells using the QiAmp DNA Blood Mini Kit (51104; QIAGEN). Subsequently, the gRNA binding region was amplified by PCR, and products were subjected to Sanger sequencing. The indel rates were analyzed using the Inference of CRISPR Edits (ICE) Analysis Tool (Synthego Performance Analysis, ICE Analysis, 2019. V3.0. Synthego). Some knockout cell lines were subjected to single-cell sorting followed by knockout verification

with PCR or TA cloning and Sanger sequencing. Three verified knockout clones were pooled in equal ratios before being used in experiments.

## CTL and apoptosis assay

Verified knockout, FLJ23584-myc-DDK plasmid-transfected or non-target control cells of the cell lines NCI-H1975 or A549 were seeded onto CELLSTAR black 96-well plates (#655090; Greiner). On the day after, these cells were stained with CellTracker Orange CMRA Dye (C34551; Thermo Fisher Scientific) for 30 min and subsequently washed with PBS. Cell numbers for each well were obtained by fluorescent image-based automated cell counting on a Celigo Imaging Cytometer (Nexcelom Bioscience). A master mix containing pre-activated T cells in an E:T ratio of 1.5:1 (NCI-H1975) or 5:1 (A549) and Her2/CD3, EpCAM/CD3 or EGFR/CD3 bispecific antibody and 0.1% Incucyte Caspase-3/7 Green Dye (4440; Sartorius) in Fluoro-Brite media (A1896701; Thermo Fisher Scientific) was added to the cells. For apoptosis assays with drugs, cells were treated with 20 µM camptothecin (C-9911; Sigma-Aldrich), 500 ng/µl nocodazole (S2775; Selleckchem), or 50 nM docetaxel (S1148; Selleckchem) diluted in DMSO (final conc. 1%) in FluoroBrite media containing caspase-3/7 fluorescent Green Dye (4440; Sartorius) followed by incubation for 18 h. Recombinant granzyme B (200 nM, AB285779; Abcam) was mixed with 50 ng/ml streptolysin O (S5265; Merck) and 1 mM DTT in FluoroBrite medium containing caspase-3/7 Green Dye (Sartorius) and incubated for 16 h. The fraction of apoptotic cells was determined by image-based cell counting of all cells (red) and apoptotic cells (green) at a Celigo imaging cytometer (Nexcelom Bioscience).

## Tagging of *C22orf46* ORF FLJ23584

For C-terminal tagging of the *C22orf46* ORF *FLJ23584* (UniProt ID Q9H5C5) a homology-directed repair donor plasmid was generated containing an expression cassette encoding eGFP coupled to a blasticidin resistance gene via a P2A sequence cloned into the vector pAL119-TK (#21911; Addgene). Homology arms complementary to the 5′ and 3′ regions of the stop codons of the *C22orf46* ORF *FLJ23584* were inserted to the tagging cassette by cloning using the Esp3I restriction sites. The plasmid DNA was transformed into the *E. coli* strain DH5α, and cells were grown overnight at 37°C with constant shaking. Plasmid DNA was purified using the QIAGEN-tip 20 Maxiprep Kit (12163; QIAGEN), and the plasmid sequences were verified by Sanger sequencing. The homology-directed repair plasmid (1 µg) along with 0.25 µl of the sgRNA (50 pmol; Synthego synthetic) targeting *FLJ23584* (5′-CTCTCCTGGCCTTTTCAGGC-3′) and 0.75 µg CleanCap Cas9 mRNA (L-7606; TriLink) were transfected into 0.2 million NCI-H1975 WT cells via 4D-Nucleofector X Unit (SF Cell Line 4D-Nucleofecto X Kit S, V4XC-2032; Lonza) according to manufacturer's protocol. Selection for blasticidin (3 µg/ml) resistant cells was started 2 wk after transfection. eGFP expression was measured via flow cytometry using MACS Quant VYB. A population of eGFP-expressing and blasticidin-resistant cells was sorted by BD FACSMelody Cell Sorter and correct integration of eGFP into the ORF *FLJ23584* was confirmed via genomic DNA isolation using the QiAmp

DNA Blood Mini Kit (51104; QIAGEN), PCR amplification, and Sanger sequencing.

## Plasmid transfection

NCI-H1975 WT cells were transfected with a myc-DDK-tagged FLJ23584 ORF clone (RC203355; Origene) using the SF Cell Line 4D-Nucleofector X Kit L (V4XC-2024; Lonza). The same plasmid was used for rescue experiments in non-targeting control and *C22orf46* knockout cell lines. For each transfection, one million cells and 5 $\mu$g plasmid were used, and the manufacturer's instructions were followed. 4 d after transfection, the selection with Geneticin (InvivoGen, ant-gn) was started. The cell populations resistant to 600 $\mu$g/ml Genetecin were subjected to downstream experiments.

## Immunofluorescence

Cells grown in eight-well $\mu$-Slides (80807; ibidi) until 80% confluency were washed with PBS and fixed in 4% PFA solution at room temperature for 30 min, followed by washing with PBS. To permeabilize cell membranes, a 0.5% Triton-X solution was applied for 15 min, and cells were washed with PBS for 10 min at 37°C. Blocking was performed using a 1% BSA solution at 37°C for 30 min, followed by another round of PBS washing. Cells were then stained with primary antibodies in a 0.5% BSA solution for 1 h at 37°C. After three washes for 5 min with PBS, cells were incubated with secondary antibodies conjugated to various fluorescent dyes for 1 h at 37°C. After three additional washes with PBS, the slides were stained with a 2.5 $\mu$g/ml DAPI solution (A1001; AppliChem) and stored in PBS. Imaging was performed using an Olympus IX71 microscope equipped with the DeltaVision Elite imaging system, using a 20×/0.95 plan apo objective. The acquired images were deconvolved and projected using softWoRx software (Applied Precision). The images in Fig S4 were obtained with Keyence BZ-X810 microscope using a 20x/0.95 plan apo objective. Finally, images were cropped, and contrast was adjusted using Fiji (Schindelin et al, 2012).

## Immunoprecipitation

Eight million eGFP-tagged NCI-H1975 cells were washed with 500 $\mu$l PBS and then incubated with 200 $\mu$l 1x RIPA lysis buffer (9806; Cell Signaling Technology) containing Halt protease and phosphatase inhibitor cocktail (1861281; Thermo Fisher Scientific) and 0.2 $\mu$l 25 U/$\mu$l benzonase (70664; Novagen) for 5 min on ice. Next, the cells were detached from the dish using a cell scraper. The resulting cell suspension was incubated on ice for 10 min, followed by ultrasonication and centrifugation at 10,000$g$ for 10 min. The supernatant was transferred to a pre-cooled tube and 300 $\mu$l dilution buffer (10 mM Tris/Cl pH 7.5, 150 mM NaCl, 0.5 mM EDTA) were added. Before use, 25 $\mu$l eGFP-Trap Magnetic beads (GTMA-20; ChromoTek) were washed twice with 500 $\mu$l ice-cold dilution buffer. The diluted lysate was added to the pre-equilibrated beads and rotated for 1 h at 4°C. Subsequently, beads were separated from the supernatant using a magnet. Beads were then resuspended in 500 $\mu$l wash buffer I (10 mM Tris/Cl pH 7.5, 150 mM NaCl, 0.05% Nonidet P40 Substitute, 0.5 mM EDTA), followed by magnetic separation and two washes with wash buffer II (10 mM Tris/Cl pH 7.5, 150 mM NaCl, 0.5 mM EDTA).

During the final wash, 10% of the sample volume was collected and prepared for Western blot analysis. For this, wash buffer II was removed, and 80 $\mu$l LDS-sample buffer was added to the beads. The sample was boiled for 5 min at 95°C to dissociate the immunocomplexes from the beads. The supernatant was then separated using a magnet and used for Western blot analysis. The remaining beads resuspended in 300 $\mu$l wash buffer II were subjected to mass spectrometry.

## Mass spectrometry

Samples were digested on-beads twice with 200 ng trypsin (V5280; Promega) for 24 h and subsequently with 50 ng rLys-C for 24 h (V1671; Promega). Digests were desalted with ultra-micro C18 columns (Nest Group), and eluates were dried in a SpeedVac (Eppendorf). For LC-MS/MS, the digests were recovered in 3 $\mu$l 30% formic acid (1.00264.0100; Merck) supplemented with 25 fmol/$\mu$l peptide and retention time standard (88321; ThermoPierce) and diluted with 20 $\mu$l HPLC-grade water (1.15333.2500; Merck). Peptide mixtures were separated by C18 reverse-phase nanoflow liquid chromatography. The LC system was directly connected to the mass spectrometer operated in data-dependent acquisition mode. Fragment spectra were extracted and submitted to automated peptide and protein identification by Mascot V2.6 (MatrixScience) software (Perkins et al, 1999). Instrumentation and software parameters are given in Table S4. Results were compiled in Scaffold V4.11 (Proteome Software) (Searle, 2010). Spectra matched to the protein sequences of interest were inspected manually.

## Flow cytometry

After detaching the cells by trypsin digest and cell counting, 200,000 cells were pelleted at 1,000$g$ for 5 min and washed with PBS. Cells were then stained with antibodies conjugated to fluorescent dyes or the corresponding isotype antibody control and incubated for 30 min at 4°C, followed by an additional 30 min at RT. Subsequently, the cells were washed with PBS, resuspended in 200 $\mu$l of PBS, and subjected to analysis on BD FACSCalibur, Canto II, MACS Quant VYB, or MACS Quant X flow cytometers. Activation status of CD8[+] T cells was verified as described previously (Sreevalsan et al, 2020) on BD LSR II or MACS Quant VYB/X flow cytometers. The obtained data were analyzed using FlowJo V10.8 Software (BD).

## Western blotting

Protein extracts were obtained from cells grown in 100 mm dishes until they reached 80% confluency. After washing with 500 $\mu$l PBS, the cells were incubated with 200 $\mu$l of 1x RIPA lysis buffer (9806; Cell Signaling Technology) containing Halt protease and phosphatase inhibitor cocktail (1861281; Thermo Fisher Scientific), along with 0.2 $\mu$l 25 U/$\mu$l benzonase (70664; Novagen) for 5 min on ice. After detachment from the dish using a cell scraper, the cell suspension was incubated on ice for 10 min, followed by ultrasonication and centrifugation at 10,000$g$ for 10 min. The resulting supernatant was used for protein quantification using the BCA assay (PierceTM BCA Protein Assay Kit #23225 and #23227) and stored at −20°C if necessary.

Appropriate amounts of protein were mixed with Nupage LDS-sample buffer (NP0008; Novex) containing 100 mM DTT. The lysates presented in Fig S4 were prepared as follows: cells were washed with PBS and trypsinated. After centrifugation at 300*g* for 5 min, the cell pellet was lysated in 1x Laemmli buffer (S3401-1VL; Sigma-Aldrich). All samples were then heated at 95°C for 5 min, followed by centrifugation at 14,000*g* for 1 min. The denatured samples were loaded onto a NuPAGE 4–12% Bis-Tris protein gel (10472322; Invitrogen) for electrophoresis, and the separated proteins were transferred to a nitrocellulose membrane. After blocking the membranes with 5% non-fat dry milk in PBS/0.1% Tween, they were incubated with the designated primary antibody in 5% non-fat dry milk in PBS/0.1% Tween overnight at 4°C. Subsequently, the membranes were washed three times for 5 min each with PBS/0.1% Tween. For secondary antibody staining, the membranes were incubated for 1 h at RT, followed by three 5-min washes with PBS/0.1% Tween. Finally, membranes were imaged using the LI-COR Odyssey imaging system; band quantification and cutting were performed using Image Studio Lite software (version 5.2.5).

### Antibodies

Immunofluorescence: rabbit anti-fibrillarin (2639; Cell Signaling Technology), donkey anti-rabbit Alexa647 (A-31573; Invitrogen), rabbit anti-DDK (TA592569; Origene), donkey anti-rabbit Alexa488 (A-21206; Invitrogen). Flow cytometry: PE mouse IgG1, κ isotype control (555749; BD PharMingen), PE mouse anti-human CD274 (557924; BD PharMingen), FITC mouse anti-human EpCAM (347197; BD PharMingen), FITC mouse IgG1, κ isotype control (349041; BD PharMingen), APC anti-human EGFR antibody (352906; BioLegend), APC mouse IgG1, κ isotype control (FC) antibody (400122; BioLegend), PE anti-human EGFR antibody (352903; BioLegend), PE mouse IgG1, κ isotype control (FC) antibody (400113; BioLegend), FITC anti-human CD340 (erb-B2/Her2) antibody (324404; BioLegend), FITC mouse IgG1, κ isotype control antibody (400108; BioLegend), FITC mouse anti-human B7-H3 (11188-MM06-F; stino biological), FITC mouse anti-human CD47 (11-0479-42; invitrogen), FITC mouse IgG1, κ isotype control antibody (11-4714-82; invitrogen). Western blot: goat anti-GAPDH (TA302944; Origene), IRDye 680LT donkey anti-goat IgG (926-68024; LI-COR Biosciences), mouse anti-eGFP (ab184601; Abcam), IRDye 800 CW donkey anti-mouse IgG (926-32213; LI-COR Biosciences), rabbit anti-DDK (TA592569; Origene), IRDye 800 CW donkey anti-rabbit IgG (926-32213; LI-COR Biosciences). Bispecific antibodies: Recombinant anti-Her2/anti-CD3, anti-EpCAM/CD3, and anti-EGFR/anti-CD3 tetravalent bispecific (scFv-hlgG1Fc-scFv)2 antibodies were cloned and produced by the DKFZ (Heidelberg).

### RNA sequencing

Triplicates of NCI-H1975 nt control and *C22orf46* knockout cell lines (gRNA 2) were cultured on six-well plates until 80% confluency. Subsequently, cells were trypsinized and washed with PBS. RNA isolation was performed according to the manufacturer's instructions using the RNeasy kit (74104; QIAGEN). Integrity of the isolated RNA samples was evaluated by agarose gel electrophoresis, and concentrations were determined using the Qubit RNA HS Assay Kit (Q32852; Invitrogen), and 500 ng of total RNA were submitted for Illumina deep sequencing. Transcript expression levels were quantified with kallisto (Bray et al, 2016) version 0.46.1 using Ensembl transcriptome release 104 (Howe et al, 2021) as a transcriptome annotation database. Differential expression analysis was performed in R (https://www.R-project.org, version 3.5) using sleuth (Pimentel et al, 2017) and tidyverse (Wickham et al, 2019) packages.

For GSEA, pre-defined gene sets (H1, CP_Biocarta & CP_KEGG) were obtained from the Molecular Signatures Database (http://www.broad.mit.edu/gsea/) (Subramanian et al, 2005). In addition, chemical and genetic perturbation gene sets containing the keywords "lung" and "cancer" were selected as a customized collection. The gene list ranked by decreasing fold change in transcript expression was submitted to Broad Institute's GSEA software version 4.1.0 using the "PreRanked" tool (parameters: number of permutations = 1,000, no collapse). Significantly enriched gene sets were defined using a nominal *P*-value < 0.01.

### qRT–PCR

NSCLC cells were cultured in six-well plates, and one million cells were used for total RNA extraction with the RNeasy kit (74104; QIAGEN). To synthesize cDNA, 500 ng of total RNA were annealed with oligo dT primers at 65°C for 5 min and the reverse transcriptases SuperScript III or IV (18080085; Thermo Fisher Scientific) were used according to the manufacturer's instructions. For quantitative real-time PCR, ABsolute QPCR Mix Sybr Green, no-ROX (AB1158; Thermo Fisher Scientific) was used with appropriate program settings. qRT-PCR primers were designed to target TATA-box-binding protein or the gene of interest. The mRNA expression levels were determined using the ΔΔCt method. The Ct values of the gene of interest were normalized to the respective Ct values for TATA-box binding protein transcript and further normalized to nt control samples. To determine transcript variants, PCR reactions on cDNA were performed with appropriate primers (Table S3), followed by Sanger sequencing.

### 3D sequence- and structure-based characterization

Sequence similarity searches were carried out with the blastp suite and the nr and pdb databases at the NCBI website (https://blast.ncbi.nlm.nih.gov). Available Alphafold predictions were obtained from the EBI website (https://alphafold.ebi.ac.uk/entry/Q9H5C5) and confirmed with in-house Alphafold (Jumper et al, 2021a, 2021b) calculations as previously reported (Guillen-Pingarron et al, 2022). Secondary structure predictions were carried out with PSIPRED 4.0 (http://bioinf.cs.ucl.ac.uk/psipred). Threading studies were performed using the Phyre[2] server (http://www.sbg.bio.ic.ac.uk/phyre2) (Kelley et al, 2015) and the I-Tasser suite (https://zhanggroup.org/I-TASSER) (Yang et al, 2015).

### Statistical analysis

Statistical analysis was carried out with GraphPad Prism (version 9.4.1). Two-way comparisons were performed by Welch's two-tailed *t*-test (degree of freedom as indicated) (*$P$-value < 0.05, **$P$-value < 0.01, ***$P$-value < 0.001). Testing for differential expression of genes

between RNA-Seq samples was performed using the standard workflow of R package DESeq2 (Love et al, 2014), with $P$-values calculated using a Wald test, followed by an FDR adjustment according to a Benjamini and Hochberg method. MFI values were calculated by building the median of the fluorescence intensities.

## Data Availability

The DeepSeq data from this publication have been deposited to the NIH Gene Expression Omnibus (GEO) database (URL: https://www.ncbi.nlm.nih.gov/geo) and assigned the identifier GSE236401.

## Supplementary Information

## Acknowledgements

This work was funded by the NCT/UCC Dresden. We thank all donors of tissue samples for the participation in this study. All donors declared their consent on the use of their tissue material for research experiments. Parts of the project were conducted under the ethics vote of the Ethics Committee of the University of Dresden (EK244072018). We thank Marten Meyer and Frank Momburg (DKFZ, Heidelberg) for providing bispecific antibodies and Marc Gentzel (TU Dresden) for mass spectrometry measurements. We thank the DRESDEN-concept Genome Center, part of the CMCB technology platform TU Dresden, for technical support. We are thankful to Gloria Ruiz-Gómez and Sandeep Sreevalsan for helpful scientific discussions.

### Author Contributions

M Döring: data curation, validation, investigation, and writing—original draft.
M Brux: validation and investigation.
M Paszkowski-Rogacz: data curation and software.
PM Guillem-Gloria: data curation, software, and investigation.
F Buchholz: data curation, funding acquisition, and writing—review and editing.
MT Pisabarro: conceptualization, data curation, software, supervision, investigation, and writing—original draft, review, and editing.
M Theis: conceptualization, data curation, supervision, funding acquisition, investigation, and writing—original draft, review, and editing.

### Conflict of Interest Statement

The authors declare that they have no conflict of interest.

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
