## [Reviewer comments · Life Science Alliance]

Life Science Alliance

Nucleolar Protein TAAP1/C22orf46 Confers Pro-Survival Signaling in Non-Small Cell Lung Cancer

Mirko Theis, Marietta Döring, Melanie Brux, Maciej Paszkowski-Rogacz, Pedro Guillem-Gloria, Frank Buchholz, and M. Pisabarro

DOI: <https://doi.org/10.26508/lsa.202302257>

Corresponding author(s): Mirko Theis, National Center for Tumor Diseases (NCT/UCC) Dresden, German Cancer Research Center (DKFZ), University Hospital Carl Gustav Carus, Technische Universität Dresden, Helmholtz-Zentrum Dresden-Rossendorf (HZDR), Dresden, Germany

Review Timeline:

Submission Date:	2023-07-07
Editorial Decision:	2023-09-08
Revision Received:	2023-12-07
Editorial Decision:	2023-12-26
Revision Received:	2024-01-04
Accepted:	2024-01-04

Transaction Report:

September 8, 2023

Re: Life Science Alliance manuscript #LSA-2023-02257

Dr. Mirko Theis

National Center for Tumor Diseases (NCT/UCC) Dresden, German Cancer Research Center (DKFZ), University Hospital Carl Gustav Carus, Technische Universität Dresden, Helmholtz-Zentrum Dresden-Rossendorf (HZDR), Dresden, Germany
NCT

Fetscherstraße 74
Dresden 01307
Germany

Dear Dr. Theis,

Thank you for submitting your manuscript entitled "Nucleolar Protein TAAP1 / C22orf46 Confers Pro-survival Signaling in Non-Small Cell Lung Cancer" to Life Science Alliance. The manuscript was assessed by expert reviewers, whose comments are appended to this letter. We invite you to submit a revised manuscript addressing the Reviewer comments.

Thank you for this interesting contribution to Life Science Alliance. We are looking forward to receiving your revised manuscript.

Sincerely,

B. MANUSCRIPT ORGANIZATION AND FORMATTING:

Reviewer #1 (Comments to the Authors (Required)):

Dear Editor, Dear Authors,

I have carefully reviewed your manuscript titled "Nucleolar Protein TAAP1 / C22orf46 Confers Pro-survival Signaling in Non-Small Cell Lung Cancer" and would like to provide you with my feedback. Overall, I believe that your study delivers an important message; however, to my opinion, there are several areas that need to be addressed before it can be considered for publication at Life Science Alliances. My comments and suggestions are outlined below:

1. While your manuscript conveys a significant finding, the data presented are currently too preliminary. I recommend conducting additional experiments to strengthen your conclusions and provide a more comprehensive understanding of the role of TAAP1/C22orf46 in tumour immune evasion.
2. The use of GFP tagging raises concerns due to its large size in comparison to the putative endogenous protein. Previous reports have indicated that GFP tags can alter protein function. To address this issue, I suggest repeating the experiments using a smaller tag, such as a flag tag or HA. Alternatively, producing an antibody to detect the endogenous protein would provide more reliable evidence.
3. If FLJ23584 contains BH domains, it is essential to investigate their functionality by assessing potential interactions with other members of the Bcl-2 protein family. This analysis would contribute to a better understanding of the molecular mechanisms underlying the pro-survival functions of TAAP1/C22orf46.
4. While the manuscript mentions that C22orf46 is expressed in many cancers, it would be valuable to investigate whether its expression is higher in cancer tissues compared to healthy tissues. Additionally, exploring its expression within different cell populations within the tumour mass, possibly by re-analyzing single-cell RNA sequencing data, could provide insights into the tumour cell-specific enrichment of this nucleolar protein.

In summary, I believe that your manuscript contains valuable findings, but additional experiments and analyses are necessary to strengthen the conclusions and fully characterize TAAP1/C22orf46. I kindly request that you address these comments in a revised version of the manuscript. Thank you for considering my feedback, and I look forward to reviewing the revised manuscript.

Reviewer #2 (Comments to the Authors (Required)):

In the present study, Döring et al. performed a serial enrichment CRISPR-Cas9 screen in non-small cell lung cancer (NSCLC) cells, to identify factors which confer these cells with the ability to evade the cytotoxic effects of CD8+ T cells engaged with bispecific antibodies. From this screen they identified the gene C22orf46 encoding the nucleolar localized protein named Tumor Apoptosis Associated Protein 1 (TAAP1) which contributes to pro-survival functions and oncogenic gene expression.

The study is of significance and the authors provide evidence to support the role of C22orf46/TAAP1 in mediating anti-apoptotic signaling in NSCLC cells. However, there are some significant concerns which I believe must be addressed to support the conclusions drawn in this manuscript.

1. In the CRISPR-Cas9 screen the authors evaluated 7 guides (C22orf46-1 - C22orf46-7) targeting the C22orf46 gene (Table S1) out of which guides C22orf46-1 and C22orf46-2 showed robust depletion only in the presence of bispecific Abs in both consecutive T cell treatments (Table S2). However, for making the knockout cell lines (Table S3) the authors used guide C22orf46-2 (gRNA1 for NCI-H1975) and guide C22orf46-1 (gRNA1 for A549) and a new guide (gRNA2 for both NCI-H1975 and

A549) which was not part of the screen and which has a large overlap with the region targeted by guide C22orf46-2 (Figure S3). The authors must provide the rationale for selection of guides used to generate the knockout cell lines utilized in the study.

2. In figure S2B, the authors evaluated the reduction of C22orf46 mRNA levels in the knockout cell lines generated using targeting guides in comparison to NT guide control samples. Potent reduction in mRNA levels is observed using gRNA1 and gRNA2 for NCI-H1975 cells. For A549 cells while gRNA2 shows significant reduction of mRNA levels, no mRNA reduction is observed for gRNA1. However in Figure 2D, an increase in apoptosis is observed for cells targeted by both these gRNAs. The authors must provide a justification for these results.

3. In order to rule out potential off-target effects contributing to the observed apoptosis instead of C22orf46 knockout, I suggest that the authors perform a rescue experiment to overexpress C22orf46 in the knockout cell lines treated with either CTL-bispecific antibodies or genotoxic agents to confirm the anti-apoptotic functions of this protein.

4. The authors endogenously tagged the C22orf46 ORFs FLJ23584 and C9J442 with EGFP using HDR templates, for evaluating their expression. The authors claim that both ORFs were correctly modified. However in Figure S3B, while for FLJ23584-eGFP, gel bands for PCR products corresponding to both left and right homology arms were observed, a clear band is not observed for the PCR product corresponding to the left homology arm for C9J442-eGFP. In the methods section, the authors mention that Sanger sequencing was done to confirm integration of the HDR template. The authors must show these results.

5. The authors must specify which knockout cell lines were used in the RNA-seq experiment. Is the data representative of knockout cell lines using both guides gRNA1 and gRNA2?

Minor correction - there is a typo in line 357 - fade instead of fate

Point-by-point response:**Reviewer #1:**

I have carefully reviewed your manuscript titled "Nucleolar Protein TAAP1/C22orf46 Confers Pro-survival Signaling in Non-Small Cell Lung Cancer" and would like to provide you with my feedback. Overall, I believe that your study delivers an important message; however, to my opinion, there are several areas that need to be addressed before it can be considered for publication at Life Science Alliances. My comments and suggestions are outlined below:

1. While your manuscript conveys a significant finding, the data presented are currently too preliminary. I recommend conducting additional experiments to strengthen your conclusions and provide a more comprehensive understanding of the role of TAAP1/C22orf46 in tumour immune evasion.

Reply:

We thank the reviewer for pointing this out. We agree that the findings are significant and that further experiments are required to corroborate the role of *C22orf46*. The revised version of the manuscript contains validation data generated by utilizing four additional knockout cell lines generated with independent sgRNAs and data on the localization of TAAP1 tagged with a myc-DDK-tag. Furthermore, we conducted an analysis of *C22orf46* transcript levels in various cancer entities compared to normal tissue using publicly available data and observed a statistically significant upregulation of *C22orf46* expression in 13 of 22 cancers including lung adeno- and sarcomatoid carcinomas (Fig 4B). Finally, we conducted experiments aiming at a better functional understanding. One possibility to deduce functional information for novel proteins is to determine interaction partners. Consequently, we conducted immunoprecipitation experiments combined with mass spectrometry. Remarkably, a number of nuclear and nucleolar proteins have been identified by this analysis fitting to the nucleolar localization of TAAP1. However, none of these proteins were significantly enriched in the IP and we have doubts whether their interaction to TAAP1 is genuine. We are convinced that additional work has to be spent on optimizing the IP protocol to gain reliable results. However, we were unable to complete this work within the revision period and feel that it may exceed the scope of revision process. In summary, the additional data fits to the conclusions drawn in the first version of the manuscript and underpin the pro-carcinogenic role of *C22orf46*.

2. The use of GFP tagging raises concerns due to its large size in comparison to the putative endogenous protein. Previous reports have

indicated that GFP tags can alter protein function. To address this issue, I suggest repeating the experiments using a smaller tag, such as a flag tag or HA. Alternatively, producing an antibody to detect the endogenous protein would provide more reliable evidence.

Reply:

We thank the reviewer for bringing this up. Indeed, eGFP tagging may alter protein function and result in misleading interpretations of experimental data. However, we like to point out that most functional conclusions were drawn from data generated in *C22orf46* knockout cell lines. Furthermore, numerous studies have successfully utilized the eGFP tag and came up with valid information on protein functions (Science, 2010, 328(5978), 593). Nevertheless, we take the reviewer's concern very serious and generated a cell line expressing myc-DDK-tagged *C22orf46*. The myc-DDK tag is composed of 25 amino acids and much smaller than the eGFP tag. Utilizing NCI-H1975 cells, we expressed TAAP1-myc-DDK from a vector and determined the intracellular localization of the tagged protein. Strikingly, we observed a nucleolar localization (Fig S4) corroborating the results obtained with TAAP1-eGFP. The data has been added to the manuscript (lines 206-208).

3. If FLJ23584 contains BH domains, it is essential to investigate their functionality by assessing potential interactions with other members of the Bcl-2 protein family. This analysis would contribute to a better understanding of the molecular mechanisms underlying the pro-survival functions of TAAP1/*C22orf46*.

Reply:

We are grateful to the reviewer for the suggestion. BH-domains interact among each other conferring apoptotic regulatory properties and binding of FLJ23584 to Bcl-2 family members is plausible. Hence, we conducted immunoprecipitation assays to identify interaction partners. These assays delivered a list of putative binding partners. However, none of these proteins harbor a BH domain. Additionally, we probed FLJ23584 for an interaction with Bcl-2 family proteins by co-expressing Bcl-2-myc-DDK, Bid-myc-DDK and Bax-myc-DDK in endogenously tagged TAAP1-eGFP NCI-H1975 cells and conducted pulldown assays utilizing an anti-eGFP antibody and an anti-myc antibody for detection. These assays did not deliver any evidence for an interaction of FLJ23584 with either Bcl-2, Bid or Bax. As BH domains are precisely adapted to their interaction partners it is well possible that FLJ23584 interacts with another BH-domain containing protein not included in our assays. Additionally, proteolytic cleavage may change the properties of BH domains as shown for Bid and tBid (Wang et al., J Biol Chem, 2013, 288(50), 35840) and may only be seen in cells with activated caspases. Consequently, we emphasize in the

manuscript that the structure of FLJ23584 is solely based on data generated by *in silico* predictions.

4. While the manuscript mentions that C22orf46 is expressed in many cancers, it would be valuable to investigate whether its expression is higher in cancer tissues compared to healthy tissues. Additionally, exploring its expression within different cell populations within the tumour mass, possibly by re-analyzing single-cell RNA sequencing data, could provide insights into the tumour cell-specific enrichment of this nucleolar protein.

Reply:

This is an important point. We analyzed *C22orf46* transcript levels in malignant versus healthy tissues utilizing TNM plot (<https://tnmplot.com>; Bartha et al., *Int. J. Mol. Sci.*, 2021, 22(5), 2622) and observed a significant upregulation of *C22orf46* expression in 13 of 22 cancer entities including lung adeno- and sarcomatoid carcinomas (Fig 4B) supporting our conclusions on the relevance of *C22orf46* in tumorigenesis. This information has been added to the revised version of the manuscript.

We are grateful to the reviewer for the suggestion to re-analysis *C22orf46* transcript levels using single-cell RNA sequencing data as this may provide a view on intratumoral heterogeneity of *C22orf46* expression. At the end of this point-by-point response, we are listing twelve single-cell RNA sequencing studies that were re-evaluated. However, most studies dismiss *C22orf46* or the transcript was detected at very low levels. Finally, we decided to refrain from including the data in the manuscript.

In summary, I believe that your manuscript contains valuable findings, but additional experiments and analyses are necessary to strengthen the conclusions and fully characterize TAAP1/C22orf46. I kindly request that you address these comments in a revised version of the manuscript. Thank you for considering my feedback, and I look forward to reviewing the revised manuscript.

Reviewer #2:

In the present study, Döring et al. performed a serial enrichment CRISPR-Cas9 screen in non-small cell lung cancer (NSCLC) cells, to identify factors, which confer these cells with the ability to evade the cytotoxic effects of CD8+ T cells engaged with bispecific antibodies. From this screen they identified the gene *C22orf46* encoding the nucleolar localized protein named Tumor Apoptosis Associated Protein 1 (TAAP1) which contributes to pro-survival functions and oncogenic gene expression.

The study is of significance and the authors provide evidence to support the role of *C22orf46*/TAAP1 in mediating anti-apoptotic signaling in NSCLC cells. However, there are some significant concerns, which I believe must

be addressed to support the conclusions drawn in this manuscript.

1. In the CRISPR-Cas9 screen the authors evaluated 7 guides (C22orf46-1 - C22orf46-7) targeting the C22orf46 gene (Table S1) out of which guides C22orf46-1 and C22orf46-2 showed robust depletion only in the presence of bispecific Abs in both consecutive T cell treatments (Table S2). However, for making the knockout cell lines (Table S3) the authors used guide C22orf46-2 (gRNA1 for NCI-H1975) and guide C22orf46-1 (gRNA1 for A549) and a new guide (gRNA2 for both NCI-H1975 and A549) which was not part of the screen and which has a large overlap with the region targeted by guide C22orf46-2 (Figure S3). The authors must provide the rationale for selection of guides used to generate the knockout cell lines utilized in the study.

Reply:

This is an important point and we apologize for not being sufficiently clear in the manuscript. In general, all sgRNAs have been designed to be specific for their target sites and to show a high Doench scores (Nat Biotechnol, 2016, 34 (2), 184). We understand the confusion of the reviewer about the utilized sgRNAs and that C22orf46-1 and -2 have been used in one of the NSCLC cell lines only. Furthermore, we agree with the reviewer in respect of the concerns using sgRNAs, which have large overlaps. We have generated the missing cell lines (C22orf46-1 for NCI-H1975; C22orf46-2 for A549), which show the same effects on sensitizing NSCLC cells for T cell/biAb or drug induced apoptosis as described for the sgRNAs presented earlier (Fig 2A,D and Fig 4C). Furthermore, we have generated cell lines in NCI-H1975 and A549 cells utilizing an additional sgRNA (sgRNA 4), which is located ~3 kb and ~6 kb distant of the sgRNAs C22orf46-1 and C22orf46-2, respectively and hence target a fully independent genomic site (Fig S2A). Knockout cells generated with sgRNA 4 show a significant increase in the susceptibility for T cell/biAb mediated lysis in both cell lines (Fig 2A,D) and with antineoplastic drugs (Fig 4C). In summary, we present *C22orf46* knockout data from four sgRNAs out of which three are targeting independent genomic loci in two NSCLC cell lines corroborating the conclusions draw in the first version of the manuscript.

Finally, we changed the nomenclature of the sgRNAs to be more concise (Table S3) and provide a schema of the genomic target sites for illustration (Fig S2A). A table of the changed sgRNA nomenclature is provided for the reviewer's convenience as follows:

First version	Revised version
C22orf46-2	sgRNA 1
C22orf46 gRNA2	sgRNA 2
C22orf46-1	sgRNA 3
---	sgRNA 4

Table R1: Nomenclature of sgRNAs comparing the first and the revised version of the manuscript.

2. In figure S2B, the authors evaluated the reduction of C22orf46 mRNA levels in the knockout cell lines generated using targeting guides in comparison to NT guide control samples. Potent reduction in mRNA levels is observed using gRNA1 and gRNA2 for NCI-H1975 cells. For A549 cells while gRNA2 shows significant reduction of mRNA levels, no mRNA reduction is observed for gRNA1. However in Figure 2D, an increase in apoptosis is observed for cells targeted by both these gRNAs. The authors must provide a justification for these results.

Reply:

We are grateful to the reviewer for raising this important point. Although frequently degraded by nonsense-mediated decay, mRNA levels are not necessarily correlating with the phenotypic strength, which is rather dictated by the loss of functional protein. Mutations in the mRNA sequence may alter RNA structure or processing and finally lead to impaired protein translation even without affecting RNA levels. Both sgRNAs are located close to a splicing site (sgRNA1_H1975: 2. exon; sgRNA1_A549: 3. exon). sgRNAs located next to splicing sites have been shown to have an increased efficacy in perturbing protein expression (García-Tuñón et al., PLoS One, 2019, 14(5): e0216674) as the introduced indel mutations interfere with RNA processing and translation even though not perturbing transcript levels.

Additionally, we would like to emphasize that the differences observed for the levels of *C22orf46* transcript in A549 and NCI-H1975 knockout cells may be attributed to difference in the sgRNA sequence, which is not identical for sgRNA 1 (NCI-H1975) and sgRNA 1 (A549). We understand that naming the sgRNAs by the cell line is misleading and apologize for the confusion. In the revised version of the manuscript, we changed the sgRNA nomenclature accordingly (Table R1).

3. In order to rule out potential off-target effects contributing to the observed apoptosis instead of C22orf46 knockout, I suggest that the authors perform a rescue experiment to overexpress C22orf46 in the knockout cell

lines treated with either CTL-bispecific antibodies or genotoxic agents to confirm the anti-apoptotic functions of this protein.

Reply:

We agree with the reviewer that off-targeting by individual sgRNAs might lead to misinterpretations and should carefully be ruled out. The revised manuscript contains data on knockout cell lines in A549 and NCI-H1975 cells generated by three sgRNAs targeting independent regions of *C22orf46*. In all *C22orf46* knockout cell lines a consistent increase of sensitivity to T cell/biAb mediated lysis and antineoplastic drugs has been detected (Fig 2A,D and Fig 4C). Furthermore, we present data on a genetic rescue experiment by exogenous expression of FLJ23584-myc-DDK in two *C22orf46* knockout cell lines (sgRNA 1 and -4) and measured apoptosis induction after treatment with antineoplastic drugs or granzyme B. In all cases, we observed that the knockout phenotype could be reversed by expression of *C22orf46* (Fig 4D). In summary, the data corroborates the conclusions raised on the function of *C22orf46* and exclude off-target effects.

4. The authors endogenously tagged the C22orf46 ORFs FLJ23584 and C9J442 with EGFP using HDR templates, for evaluating their expression. The authors claim that both ORFs were correctly modified. However in Figure S3B, while for FLJ23584-eGFP, gel bands for PCR products corresponding to both left and right homology arms were observed, a clear band is not observed for the PCR product corresponding to the left homology arm for C9J442-eGFP. In the methods section, the authors mention that Sanger sequencing was done to confirm integration of the HDR template. The authors must show these results.

Reply:

We apologize for not being clear. The Sanger sequencing results of the right site of C9J442 and both sites of FLJ23584 have been provided along with the manuscript (Table S3, spreadsheet D). However, the sequence verification of the left integration site in C9J442 is missing, as we fail in PCR amplification. Although this does not disprove proper genomic integration per se, we decided to remove all data based on C9J442 tagging from the manuscript. However, as the vast majority of transcripts in human are monocistronic, the conclusions raised on FLJ23584 remain unaffected.

5. The authors must specify which knockout cell lines were used in the RNA-seq experiment. Is the data representative of knockout cell lines using both guides gRNA1 and gRNA2?

Reply:

We apologize for not providing this information in the first version of the manuscript. The RNA-seq experiments have been conducted utilizing the *C22orf46* knockout cells generated with the sgRNA 2 (new nomenclature). This information has been added to the manuscript in line 642. As the experiments were conducted with those knockout cells, we are unable to provide information on how representative the data is compared to *C22orf46* knockouts generated by other sgRNAs.

Minor correction - there is a typo in line 357 - fade instead of fate.

Reply:

Many thanks for spotting this typo. It has been corrected.

Appendix: Single-Cell Expression Data on *C22orf46*

Single-cell RNA sequencing studies examined:

1. Bischoff et al., *Oncogene*, 2021, 40, 6748-6758
 - *C22orf46* is not present in this data set.
2. Zhang et al., *Signal Transduction. and Targeted Therapy*, 2022, 7, 9
 - *C22orf46* expression very low.
3. Ravi et al., *Nature Genetics*, 2023, 55, 807–819
 - Raw data is not accessible.
4. Goveia et al., *Cancer Cell*, 2020, 37, 21-36
 - No correlation between *C22orf46* expression and distinct populations of cells.
5. Dong et al., *Proteomics and Systems Biology*, 2020, 20, 13
 - *C22orf46* sparsely detected and uniformly expressed in all cell clusters.

Single Cell Portal - Broad Institute Studies:

1. https://singlecell.broadinstitute.org/single_cell/study/SCP109/melanoma-immunotherapy-resistance?genes=C22orf46&tab=scatter#study-visualize
2. https://singlecell.broadinstitute.org/single_cell/study/SCP11/melanoma-intra-tumor-heterogeneity?genes=C22orf46&tab=scatter#study-visualize
3. https://singlecell.broadinstitute.org/single_cell/study/SCP1332/genomic-and-transcriptomic-correlates-of-immunotherapy-response-within-the-tumor-microenvironment-of-leptomeningeal-metastases?genes=C22orf46&tab=scatter#study-visualize
4. https://singlecell.broadinstitute.org/single_cell/study/SCP1415/cryopreservation-of-human-cancers-conserves-tumour-heterogeneity-for-single-cell-multi-omics-analysis?genes=C22orf46#study-visualize
5. https://singlecell.broadinstitute.org/single_cell/study/SCP1064/multi-modal-pooled-perturb-cite-seq-screens-in-patient-models-define-novel-mechanisms-of-cancer-immune-evasion?genes=C22orf46&tab=scatter#study-visualize
6. https://singlecell.broadinstitute.org/single_cell/study/SCP1096/human-treated-pdac-snuc-seq?genes=C22orf46&tab=distribution#study-summary
 - 1-6: No correlation between *C22orf46* expression and distinct cell populations.
7. https://singlecell.broadinstitute.org/single_cell/study/SCP1106/stromal-cell-diversity-associated-with-immune-evasion-in-human-triple-negative-breast-cancer?genes=C22orf46#study-visualize

8. https://singlecell.broadinstitute.org/single_cell/study/SCP398/defining-t-cell-states-associated-with-response-to-checkpoint-immunotherapy-in-melanoma?genes=C22orf46&tab=scatter#study-visualize
9. https://singlecell.broadinstitute.org/single_cell/study/SCP1106/stromal-cell-diversity-associated-with-immune-evasion-in-human-triple-negative-breast-cancer?genes=C22orf46#study-visualize
10. https://singlecell.broadinstitute.org/single_cell/study/SCP1162/human-colon-cancer-atlas-c295?genes=C22orf46#study-visualize
 - 7-10: Only cells in the tumor environment considered, no data on tumor cells.
11. https://singlecell.broadinstitute.org/single_cell/study/SCP739/single-cell-transcriptomics-of-human-and-mouse-lung-cancers-reveals-conserved-myeloid-populations-across-individuals-and-species?genes=C22orf46&tab=scatter#study-visualize
12. https://singlecell.broadinstitute.org/single_cell/study/SCP1244/transcriptional-mediators-of-treatment-resistance-in-lethal-prostate-cancer?genes=C22orf46#study-visualize
 - 11-12: *C22orf46* expression very low or not detected.

December 26, 2023

RE: Life Science Alliance Manuscript #LSA-2023-02257R

Dr. Mirko Theis

National Center for Tumor Diseases (NCT/UCC) Dresden, German Cancer Research Center (DKFZ), University Hospital Carl Gustav Carus, Technische Universität Dresden, Helmholtz-Zentrum Dresden-Rossendorf (HZDR), Dresden, Germany
NCT

Fetscherstraße 74
Dresden 01307
Germany

Dear Dr. Theis,

Thank you for submitting your revised manuscript entitled "Nucleolar Protein TAAP1/C22orf46 Confers Pro-Survival Signaling in Non-Small Cell Lung Cancer". We would be happy to publish your paper in Life Science Alliance pending final revisions necessary to meet our formatting guidelines.

- please add a Running Title to our system
- please add the Twitter handle of your host institute/organization as well as your own or/and one of the authors in our system
- please remove Character Count from the title page
- author Frank Buchholz's selected contributions do not qualify a contributor for authorship. Please either update the contributions in our system and in the Author Contributions section of the manuscript, or let us know if the author should be removed.

A. FINAL FILES:

B. MANUSCRIPT ORGANIZATION AND FORMATTING:

Sincerely,

Reviewer #1 (Comments to the Authors (Required)):

Authors replied to my concerns. I propose to accept the current version of the manuscript.

Reviewer #2 (Comments to the Authors (Required)):

The authors have responded well to the reviews and addressed all of my major concerns. The clarifications and new data provided in the revised version have enhanced the quality of the manuscript. As such, I would recommend acceptance of the revised manuscript for publication.

January 4, 2024

RE: Life Science Alliance Manuscript #LSA-2023-02257RR

Dr. Mirko Theis

National Center for Tumor Diseases (NCT/UCC) Dresden, German Cancer Research Center (DKFZ), University Hospital Carl Gustav Carus, Technische Universität Dresden, Helmholtz-Zentrum Dresden-Rossendorf (HZDR), Dresden, Germany

NCT

Fetscherstraße 74

Dresden 01307

Germany

Dear Dr. Theis,

Thank you for submitting your Research Article entitled "Nucleolar Protein TAAP1/C22orf46 Confers Pro-Survival Signaling in Non-Small Cell Lung Cancer". It is a pleasure to let you know that your manuscript is now accepted for publication in Life Science Alliance. Congratulations on this interesting work.

DISTRIBUTION OF MATERIALS:

Again, congratulations on a very nice paper. I hope you found the review process to be constructive and are pleased with how the manuscript was handled editorially. We look forward to future exciting submissions from your lab.

Sincerely,

Eric Sawey, PhD

Executive Editor

Life Science Alliance

<http://www.lsajournal.org>